# Early stimulated immune responses predict clinical disease severity in hospitalized COVID-19 patients

Rebecka Svanberg [1], Cameron MacPherson[2], Adrian Zucco [2], Rudi Agius[1], Tereza Faitova[1],
Michael Asger Andersen[1], Caspar da Cunha-Bang[1], Lars Klingen Gjærde [1], Maria Elizabeth Engel Møller[3],
Patrick Terrence Brooks[3], Birgitte Lindegaard[4,5], Adin Sejdic[4,6], Zitta Barrella Harboe[4,5], Anne Ortved Gang[1,5,7],
Ditte Stampe Hersby[1,7], Christian Brieghel [1,7], Susanne Dam Nielsen[5,8], Daria Podlekareva[9],
Annemette Hald [8], Jakob Thaning Bay[3], Hanne Marquart [3], Jens Lundgren [2,5,8], Anne-Mette Lebech[5,8],
Marie Helleberg[2,8], Carsten Utoft Niemann [1,5,10] & Sisse Rye Ostrowski [3,5,10] ✉

## Abstract

**Background** The immune pathogenesis underlying the diverse clinical course of COVID-19 is poorly understood. Currently, there is an unmet need in daily clinical practice for early biomarkers and improved risk stratification tools to help identify and monitor COVID-19 patients at risk of severe disease.

**Methods** We performed longitudinal assessment of stimulated immune responses in 30 patients hospitalized with COVID-19. We used the TruCulture whole-blood ligand-stimulation assay applying standardized stimuli to activate distinct immune pathways, allowing quantification of cytokine responses. We further characterized immune cell subsets by flow cytometry and used this deep immunophenotyping data to map the course of clinical disease within and between patients.

**Results** Here we demonstrate impairments in innate immune response pathways at time of COVID-19 hospitalization that are associated with the development of severe disease. We show that these impairments are transient in those discharged from hospital, as illustrated by functional and cellular immune reconstitution. Specifically, we identify lower levels of LPS-stimulated IL-1β, and R848-stimulated IL-12 and IL-17A, at hospital admission to be significantly associated with increasing COVID-19 disease severity during hospitalization. Furthermore, we propose a stimulated immune response signature for predicting risk of developing severe or critical COVID-19 disease at time of hospitalization, to validate in larger cohorts.

**Conclusions** We identify early impairments in innate immune responses that are associated with subsequent COVID-19 disease severity. Our findings provide basis for early identification of patients at risk of severe disease which may have significant implications for the early management of patients hospitalized with COVID-19.

### Plain language summary

The manifestation of COVID-19 varies from asymptomatic to severe pneumonia requiring ventilator support or multi-organ failure. It is still poorly understood how the immune response to SARS-CoV-2 affects the development of severe disease. There is also a lack of clinical tools to identify patients early with high risk of poor outcome. Here, we looked for potential markers of disease severity in hospitalized COVID-19 patients using method to measure individual immune reactions. We found associations between impaired immune response pathways at time of hospitalization and development of severe and critical COVID-19 disease, and we identified a number of immune markers that could be used to predict poor outcome. Our findings could help identify at-risk patients upon hospitalization, enabling closer monitoring and earlier interventions.

A full list of author affiliations appears at the end of the paper.

The coronavirus disease 2019 (COVID-19) pandemic caused by severe acute respiratory syndrome coronavirus 2 (SARS-CoV-2) remains a global health crisis, having already claimed over 6 million lives by January 2022[1,2]. The clinical presentation and disease course of COVID-19 is heterogenous, varying from asymptomatic or mild symptoms to severe pneumonia with acute respiratory distress syndrome (ARDS) requiring mechanical ventilation (MV) or septic shock with multi-organ failure[3,4]. Severe symptoms usually develop within 1–2 weeks after symptom onset[5,6]. During the first pandemic waves, approximately 15% of SARS-CoV-2 PCR-positive cases developed the severe disease, and 5% required intensive care and/or MV[7–10]. While the emergence of vaccines has remarkably improved these outcomes[11–13], hospitalization due to COVID-19 still entails risk for critical disease and death, especially among patients who are unvaccinated or have insufficient or declining vaccine response[14,15]. Risk factors for severe disease and death among both unvaccinated and vaccinated patients include older age, male gender, and pre-existing comorbidities such as obesity, hypertension, and type 2 diabetes, as well as conditions associated with immunosuppression[16,17]. Despite improvements in disease-related outcomes, COVID-19 still challenges health care systems worldwide, warranting means for upfront risk stratification of patients at the time of admission.

The immunological mechanisms underlying the diverse clinical presentation and course of COVID-19 are still poorly understood. Early studies reported that patients hospitalized with COVID-19 display neutrophilia, eosinopenia, and lymphocytopenia alongside systemic elevation of routine inflammatory markers such as C-reactive protein, ferritin, and D-dimer[6,18]. These factors are even more pronounced in severe disease[4,6]. Studies further indicate that activation of an uncontrolled systemic inflammatory response, characterized by the release of pro-inflammatory cytokines, a so-called "cytokine storm", is a key mechanism contributing to the development of critical illness and ARDS[19–21]. Coherently, numerous studies highlight the presence of elevated circulating/plasma pro-inflammatory cytokines in patients with severe COVID-19[22–25], and high levels of interleukin (IL)−6 and tumor necrosis factor (TNF)-α in early disease correlate with severe disease trajectory and increased mortality[22]. Type I and III interferon (IFN) responses have also been shown to play an important role in the pathogenesis underlying severe COVID-19, and several studies highlight disturbances in the complex regulation of type I IFNs in different anatomical compartments as well as various stages of disease development[24–30]. Correspondingly, the presence of autoantibodies against type I IFNs were found enriched in patients with critical disease[31]. Further immune characterization studies have revealed dysregulation of the myeloid immune cell compartment associated with severe COVID-19, including evidence of emergency myelopoiesis with neutrophil precursors, and downregulation of HLA-DR on monocytes[25,32,33]. Thus, the innate immune cascades occurring in early disease likely play a determining role in subsequent disease severity trajectories.

There is an unmet need in daily clinical practice for implementable biomarkers and improved risk stratification at the time of hospital admission that can help identify patients at risk of severe or critical clinical course. The present study investigated early as well as temporal changes in immune function in patients hospitalized with COVID-19, hypothesizing that functional impairments in immunity precedes and predicts severe COVID-19 disease course. In a prospective clinical study, we conducted longitudinal assessment of immune function in 30 patients hospitalized with COVID-19 in Denmark using TruCulture, a clinically implemented, commercially available whole-blood ligand-stimulation assay. TruCulture applies standardized stimuli to activate distinct immune-response pathways, including Toll-like receptors (TLRs) 4, 7, and 8, and T cell receptor (TCR) signaling, allowing quantification of cytokine- and chemokine responses[34]. Together with immune cell subsets characterization by flow cytometry, patient characteristics/demographics, and mapping of clinical disease courses, we assessed stimulated immune responses at the time of hospitalization, during the disease course, and upon discharge. We identified early functional impairments in innate immune responses at the time of hospitalization that were associated with subsequent COVID-19 disease severity, and observed reconstitution of these impairments in recovering patients. Furthermore, we identified functional biomarkers constituting an early immune response signature for predicting disease severity at the time of hospitalization.

## Methods

**The COVIMUN study**. The COVIMUN study is a prospective study of patients hospitalized due to SARS-CoV-2 confirmed infection in the Capital Region, Copenhagen, Denmark. Thirty patients enrolled in the COVIMUN study at three hospitals were included in the original cohort used for this study, and a validation cohort of twenty patients were selected for validation of our immune response signature. Patients in the original cohort were enrolled between April 19th 2020 and October 14th 2020, and patients in the validation cohort were enrolled between October 14th 2020 and January 3rd 2021. Inclusion criteria were (1) confirmed SARS-CoV-2 infection (PCR), (2) hospital admission due to COVID-19, (3) written informed consent to inclusion in the study. Exclusion criteria were the absence of any of the inclusion criteria. All patients gave written informed consent before inclusion; the study was conducted in accordance with the Declaration of Helsinki. The study was conducted under approval by the Ethical Committee (H-20026502) and Data Protection Agency (P-2020-426). Fresh blood samples were obtained at time of enrollment, day three, day seven, and weekly thereafter during hospital admission until discharge or death. Additional samples were collected at admission to the ICU. All blood samples were analyzed immediately. All measurements presented in this study came from distinct samples.

**Clinical data**. Patient demographics, clinical data on co-existing diagnoses, previous and ongoing medications or treatments (e.g., chemotherapy regimens), administered medication during hospitalization (e.g., dexamethasone), and data for mapping clinical disease severity trajectories were obtained from electronic health records. We followed patients from the time of hospital admission to either death during hospitalization or discharge, and follow-up with final outcomes were available for all patients in the cohort. Day 0 was defined as the day of hospitalization due to suspected or confirmed COVID-19, since a subset of patients had a confirmed SARS-CoV-2 test after admission. Two patients were hospitalized for other causes prior to infection. For these patients, day 0 was defined as the day of the transfer to a COVID-19 ward or infectious disease department. Clinical disease trajectories were mapped out on a day-to-day basis for each patient based on clinical data that included time of symptom onset, time of positive SARS-CoV-2 test, time of hospital admission (or time of transfer to a COVID-19 ward or equivalent for patients already hospitalized due to other causes), daily vital parameters and supplementary oxygen needed, time of admission to an intensive care unit, need for mechanical ventilation, and discharge or death (during hospitalization).

The definition of a pre-existing condition associated with immune suppression is described in Supplementary Methods.

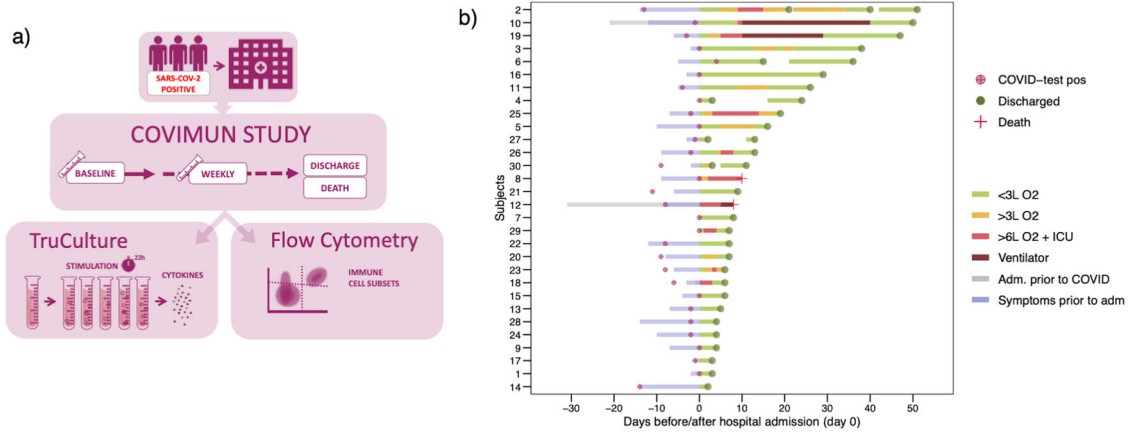

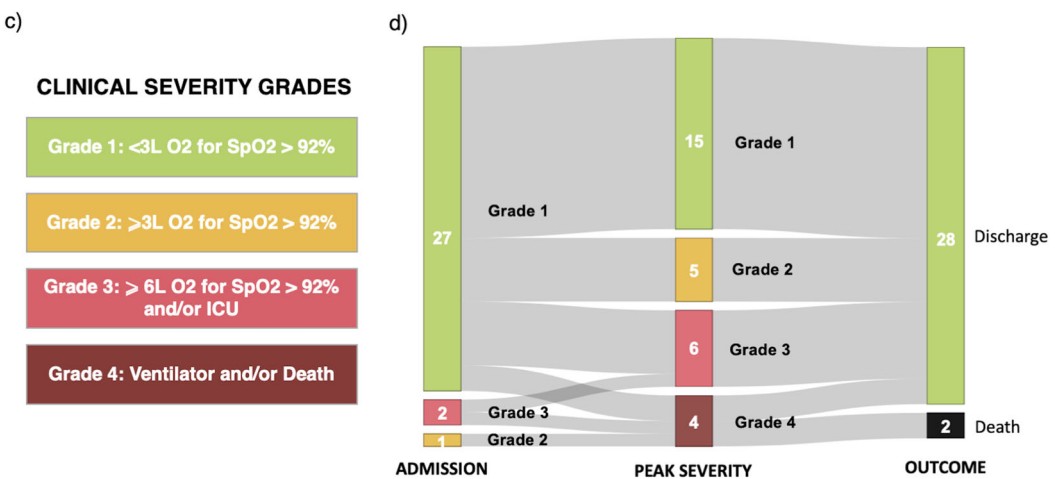

**Fig. 1 The COVIMUN study design and clinical disease severity trajectories of patients hospitalized with COVID-19. a** Flow chart of the COVIMUN study setup. Patients hospitalized with COVID-19 were included in the study. Blood samples for immediate analysis by TruCulture and Flow Cytometry were collected upon hospital admission (baseline), day three, seven, and hereafter weekly until discharge or death. **b** The clinical disease trajectories of all patients in this cohort ($n = 30$). Day 0 represents the day of hospitalization due to COVID-19. The time of positive SARS-CoV-2 PCR test, and time of discharge or death are shown. Blue bars illustrate days with symptoms prior to hospitalization. Two patients were hospitalized for other reasons prior to COVID-19 diagnosis (subjects 10 and 12, gray bars). Daily mapping of disease severity during hospital admission are illustrated by the bar colors. **c** Visual representation of a clinical severity scale defining four grades of disease severity applied throughout this study. **d** Clinical disease severity grade of all patients ($n = 30$) at time of hospital admission (admission, day 0), corresponding peak severity grade during hospitalization (peak severity), and corresponding outcome of discharge or death (outcome). COVID-19, coronavirus disease 2019; L O2, liters/minute of oxygen supply; ICU, intensive care unit; SpO2, peripheral blood oxygen saturation.

**Defining a clinical severity scale, grouping of patients, and timepoints for comparison**. Based on the clinical disease trajectories, we defined a clinical severity scale with 4 grades of disease severity (Fig. 1c), modified from a previously published COVID-19 severity grading system[24]. Correspondence between the clinical severity scale and the World Health Organization (WHO) COVID-19 disease severity classifications[35] is described in Supplementary Methods. Based on these trajectories, we decided on three timepoints for which patients were aligned for comparison, and selected the best representative samples. For "Baseline", we selected the first blood sample taken after hospital admission. In addition, the "Baseline" sample had to be collected within the first seven days of admission, and for patients with short admissions, the sample needed to be collected closer to admission than discharge. Baseline samples were thus missing for seven patients. For "Peak Severity", we selected the sample estimated to be collected closest to each patients' peak clinical disease severity. For "Discharge", we selected the sample closest to the time of discharge, which thus, only included surviving patients. Additionally, for

short admissions, the "Discharge" samples needed to be closer to discharge than to admission. At "Baseline" and "Discharge", patients were grouped based on the peak clinical severity grade of their disease trajectories (Supplementary Fig 1a, c). Patients with death as a final outcome were grouped together with severity Grade 4 (at "Baseline" only). At "Peak Severity", patients were grouped based on their *current* clinical severity grade at time of sample collection. Handling of missing samples and thus, missing data, at each timepoint is described in Supplementary Methods.

**Whole blood stimulated cytokine response assay/TruCulture**. Blood was sampled in lithium heparin tubes and transferred immediately to the laboratory. In brief, 1 h (±15 min) after blood sampling, 1 ml of whole blood was aliquoted to each prewarmed TruCulture tube (Myriad RBM, Austin, TX, US), and inserted into a digital dry block heater (WWR International A/S, Søborg, Denmark) and maintained at 37 °C for 22 h (±30 min), according to the manufacturer's recommendations. At the end of the incubation period, TruCulture tubes were opened and a valve was

inserted in order to separate the sedimented cells from the supernatant and to stop the stimulation reaction. Liquid supernatants were aliquoted and immediately frozen at −20 °C (and transferred to −80 °C after 1–7 days) until use. The TruCulture panel comprised 5 different tubes/immune cell stimuli: Bacterial endotoxin (lipopolysaccharide, LPS) from E.coli, O111:B4 providing immune cell stimulation through TLR4; Resiquimod R848 (R848, imidazoquinoline compound, and potent TLR7/8 stimulator) providing stimulation through TLR7/TLR8 mimicking (viral) single-stranded RNA (ssRNA); Polyinosinic:polycytidylic acid (Poly I:C), synthetic double-stranded RNA (dsRNA) analog) providing stimulation through TLR3 mimicking (viral) dsRNA; anti-CD3/CD28 (providing T cell stimulation through the T cell receptor and co-stimulatory receptor CD28) and a blank (Null) containing TruCulture media without stimuli, revealing in vivo blood immune cell activation and a proxy for circulating plasma levels. The concentration of IFN-α, IFN-γ, IL-1β, IL-6, IL-8, IL-10, IL-12p40, IL-17A, and TNF-α was measured in each liquid supernatant by a 9-plex Luminex (R&D Systems, BIO-Techne LTD, Abingdon, UK) using a Luminex 200 instrument (LX200, R&D Systems, BIO-Techne LTD, Abingdon, UK), according to the manufacturers recommendations. Results are reported in pg/ml. Reference intervals for all cytokine levels from all stimuli were based on TruCulture data from 31 healthy individuals and represent the range between the minimum and maximum cytokine concentration levels measured for each cytokine/stimulus.

**Flow cytometry/DuraClone**. Parallel analyses of immune cell subset concentrations by whole-blood flow cytometry were performed for most fresh samples collected in the study. All analyses were performed on fresh samples within 24 h of collection. A special designed 10-color flowcytometry panel (DuraClone) with prefabricated dried antibodies from Beckman Coulter was used (Supplementary Table 1). The antibodies were tested and titrated on normal blood cells by the manufacturer and saturated concentrations of antibody were added to the dry, unitized antibody panels. The tube contained beads for the calculation of absolute concentrations of linage populations. It was a part of a larger panel customized for primary and secondary immunodeficiency evaluation of leukocyte subsets (ref PMID: 32979342). EDTA blood was drawn from patients and prepared according to the manufacture's guidelines. All samples were analyzed on a Navios Ex flowcytometer from Beckman Coulter and results were analyzed in Kaluza Analysis 2.1 from Beckman Coulter. The gating strategies for the immune cell subset populations are outlined in Supplementary Table 2.

**Statistical analyses and data visualization**. Data visualization and statistical analyses were performed using R software version 4.0.3[36]. The swimmerplot, boxplots, dotplots, corrplots, and visualization of principal component analyses (PCA) were created using the *ggplot2* package[37]. The Sankey plots were created using the *networkD3* package version 0.4[38]. TruCulture cytokine data and DuraClone immune cell subset data exhibited a log-normal distribution, therefore these data were log-transformed for all analyses performed.

Distribution of TruCulture cytokine concentrations and DuraClone immune cell subset counts between peak severity groups at baseline and at/near peak severity were compared using the Kruskal-Wallis test with adjustment for multiple testing using Bonferroni (9 tests for TruCulture data; 9 cytokines within one stimulus, 8 tests for DuraClone data; 8 immune cell subsets). Post-hoc Dunn's test was performed where the Kruskal-Wallis test was statistically significant after adjustment for multiplicity, also using Bonferroni to adjust for multiple testing (adjustment

for 6 tests; Grade 1 vs 2, Grade 1 vs 3, Grade 1 vs 4, Grade 2 vs 3, Grade 2 vs 4, and Grade 3 vs 4). Adjusted $p < 0.05$ was considered statistically significant. Summary statistics from all Kruskal-Wallis tests at baseline and at/near peak severity are presented in Supplementary Tables 3–6. TruCulture cytokine concentrations and DuraClone immune cell subset counts at baseline vs discharge were compared using the Wilcoxon signed-rank test, only including patients with paired baseline- and discharge samples available ($n = 18$ for TruCulture, $n = 8$ for DuraClone). Adjustment for multiple testing was done using Bonferroni (9 tests for TruCulture data; 9 cytokines within one stimulus, 8 tests for DuraClone data; 8 immune cell subsets), and adjusted $p < 0.05$ was considered statistically significant. Summary statistics from all Wilcoxon signed-rank tests are presented in Supplementary Tables 7 and 8.

Pearson correlation matrices with cytokine concentrations and immune cell subset count at baseline, at/near peak severity, and at discharge were visualized using the *corrplot* package[39]. All data were log-transformed and clustered using hierarchical cluster under default parameters (complete linkage). *P*-values for each pair of variables at each timepoint were calculated and adjusted for multiple testing using Bonferroni, adjusted $p < 0.05$ was considered statistically significant.

Associations between TruCulture cytokine concentrations and peak severity grade were tested by ordinary least squares (OLS) adjusting for age as follows:

$$\text{Severity} = \log(\text{Cytokine levels}) + \text{Age}$$

Where "Severity" is the peak severity grade indicated ordinally with values 1–4, and "Age" is the age of each patient at the time of inclusion in the study. We also tested associations between "Severity" and each of the following covariates: "Age", "Sex" (sex at birth categorically encoded as 0 and 1 for females and males respectively), "Days in hospital" (representing days from hospitalization to time of collecting the sample), "Immune suppression" (binary indication of whether an immunosuppressive pre-condition was present encoded 0 and 1 for yes and no respectively), and "Admission hospital" (indicating which of the three hospitals the patient was admitted to, each hospital as a variable, encoded 1 and 0 for yes and no respectively). No statistically significant associations were identified; however, "Age" was the variable with the strongest non-significant association ($p = 0.08$). Because of this, and given that age is a well-established major risk factor for severe COVID-19[16], we chose to nevertheless adjust for "Age" in the OLS, while omitting the other covariates. The obtained p-values were adjusted for multiple tests using Bonferroni (adjustment for 45 tests; 9 cytokines × 5 stimuli). Associations were considered statistically significant if the Bonferroni adjusted p-value was below 0.05. Regression slopes and regression coefficient estimates were plotted with 95% confidence intervals using the *jtools* package version 2.1.0[40]. Summary statistics from all OLS at baseline, at/near peak severity, and at discharge are presented in Supplementary Tables 9, 10, and 11 respectively.

PCA were performed using R software version 4.0.3[36] or using Qlucore Omics Explorer software (Qlucore AB, Lund, Sweden). Qlucore Omics Explorer software (Qlucore AB, Lund, Sweden) was also used to perform the Isomap analysis. Using PCA and Isomap analyses, we examined the correlation structure across all variables and then subsets of stimuli, as defined elsewhere. PCA did not show clear separation of severity levels (Supplementary Fig. 8a), however isometric feature mapping (Isomap)[41] showed at least one manifold with a monotonic relationship to severity.

We applied OLS in combination with LASSO penalties to find the subset of predictive variables that were most likely to generalize our model, with our relatively small sample size and

large number of variables being a core motivation for this approach, using the *glmnet*-package[42]. The outcome was Severity, where the peak severity grade was indicated ordinally with values 1-4, the predictors represented the subset of baseline variables included in each bin tested (specified below), α was specified as 1 (indicating pure lasso regression), and nfolds was set to 23 (the number of samples). Standardized log-transformed cytokine concentrations were used for all analyses. Baseline variables were assessed in bins of stimuli to limit potential for overfitting. These bins were: (i) each stimulus individually, (ii) combinations of two and three ("no stimulation" and CD3/CD28 were not tested in combinations due to very low and no signal respectively), (iii) all stimuli together with exception to CD3/CD28 whose response cytokines showed no signal, and (iv) combining only the three significant variables from the individual OLS analyses (LPS-stimulated IL-1β, R848-stimulated IL-12 and IL-17A). All bins were tested with and without including "Age" as a variable (Supplementary Fig. 8g, h). For each bin, the best lambda (Supplementary Fig. 8d) was used to fit a penalized regression model and extract the coefficients for the best model, where coefficients >0 or <0 were considered of importance for the model (thus included in model), and coefficients = 0 were insignificant (thus excluded from the model). Predictions were then performed on the original cohort using the subset of variables on which the best model was based. $r^2$ (coefficient of determination) for each bin (Supplementary Fig. 8c) was calculated as follows:

$$r^2 = 1 - (\textstyle\sum(\text{Severity\_Predicted-Severity})^2 / \textstyle\sum(\text{Severity-mean(Severity)})^2)$$

Where "Severity" is the peak severity grade indicated categorically with values 1–4, and "Severity_Predicted" is the predicted peak severity based on the given model. For validation of the best model, predictions were performed based on baseline TruCulture cytokine concentration data (standardized and log-transformed) from the validation cohort ($n = 20$). We predefined the thresholds for categorizing the (continuous) model output as follows; <1.5: Severity group 1, ≥1.5 < 2.5: Severity group 2, ≥2.5 < 3.5: Severity group 3, and ≥3.5: Severity group 4. Sensitivity/recall, specificity, false positive rate, false negative rate, positive predictive value/precision, negative predictive value, false discovery rate, false omission rate, positive likelihood ratio, negative likelihood ratio, diagnostic odds ratio, and Mathew's correlation coefficient were calculated for predicting severity grade 3 or 4 vs grade 1 or 2 and vice versa, as well as severity grade 1 alone and severity grade 4 alone (Supplementary Fig. 10a).

Heatmaps were created using the *ComplexHeatmaps* package[43]. Standardized log(cytokine concentrations) were depicted on heatmaps, the x-axes represent patients grouped by peak severity, and the y-axes represent the cytokine variables grouped by inclusion/exclusion in the corresponding lasso regression model. Hierarchical clustering using Euclidean distance as dissimilarity metric was performed within groups/splits.

**Reporting summary**. Further information on research design is available in the Nature Research Reporting Summary linked to this article.

## Results

**Patient characteristics, disease severity, and clinical outcome.** Thirty patients were included in this study based on the criteria (1) PCR-confirmed SARS-CoV-2 infection, (2) hospitalization due to COVID-19, and (3) informed consent to the study. The median age was 70 years, and two thirds of the patients were male. The median body mass index was 24.5 kg/m² (range 15–43). Baseline characteristics are summarized in Supplementary Table 12. The study design is illustrated in Fig. 1a. Patients

were hospitalized for a median of 8 days (range 2–50). The median number of days with symptoms prior to hospitalization was 6 (range 0–14). Clinical severity during disease trajectories were mapped for each patient based on a day-by-day monitoring of levels of oxygen needed, need for treatment in the intensive care unit (ICU), and need for MV (Fig. 1b–d). We defined 4 grades of disease severity (Fig. 1c), modified from a previously published COVID-19 severity grading system[24]. The grading system differs from the World Health Organization's (WHO) severity classification[35] by being based primarily on interventions needed rather than diagnostic work-up/criteria (correspondence between the two grading systems is described in Supplementary Methods). Most patients were at grade 1 at the time of hospital admission and upon baseline sample collection (Fig. 1d and Supplementary Fig. 1a). Fifteen patients (50%) remained at grade 1 throughout hospitalization, five patients (17%) reached peak severity grade 2, six patients (20%) reached peak severity grade 3, and four patients (13%) reached peak severity grade 4 (Fig. 1d). The in-hospital mortality was 7% as 28 patients were discharged alive (none to palliation) and two died during hospitalization (Fig. 1d). Based on these disease trajectories, we defined time of admission (baseline), time of peak severity, and time of discharge for surviving patients for comparisons of samples collected (corresponding disease severity levels are shown in Supplementary Fig. 1a–c). Seventeen patients (57%) had conditions associated with immunosuppression at time of hospitalization, including severe multimorbidity, acquired immunodeficiency syndrome (AIDS), recent chemotherapy treatment, or ongoing immunosuppressive treatment (outlined in detail in Supplementary Methods and presented in Supplementary Table 13). These patients were proportionally evenly distributed across the four peak severity groups (Supplementary Fig. 1d). Twenty-six patients (87%) had at least one comorbidity, and eleven patients (37%) had two or more comorbidities. Comorbidities were also proportionally evenly distributed across peak severity groups (Supplementary Fig. 1e, f). Dexamethasone treatment was not approved as standard of care at the time of the pandemic where most patients in our study were included[44], therefore only nine patients received dexamethasone throughout the study of which six had it administered prior to baseline sampling. They were also proportionally evenly distributed across peak severity groups (Supplementary Table 13).

**Stimulated immune response at baseline reflects subsequent disease severity.** TruCulture whole blood immune responses were assessed by measuring levels of IFN-α, IFN-γ, IL-1β, IL-6, IL-8, IL-10, IL-12p40 (referred to as IL-12 in this study), IL-17A, and TNF-α, in response to five distinct stimuli: (1) the bacterial endotoxin lipopolysaccharide (LPS, TLR4 agonist), (2) single stranded RNA-virus analog resiquimod (R848, TLR7/8 agonist), (3) CD3/CD28 TCR/co-receptor stimulation, (4) Poly-inosinic:polycytidylic acid (Poly I:C, double-stranded RNA virus analog, TLR3 agonist) and (5) no stimulation. Baseline immune responses were based on data from the first blood sample drawn after hospital admission ($n = 23$, baseline samples missing for seven patients due to late inclusion in the study). Baseline data were compared by grouping patients based on their peak disease severity grade reached during hospitalization (Fig. 2a, Supplementary Fig. 2a–e and Supplementary Table 3), with death classified as part of grade 4 (Fig. 1d). Corresponding current severity grade at time of baseline sample collection is shown in Supplementary Fig. 1a. For the unstimulated immune responses at baseline, we observed elevated levels of IL-6 for all groups, and elevated IL-8 for the two lowest severity groups, compared to the normal reference levels. Levels of the remaining cytokines were

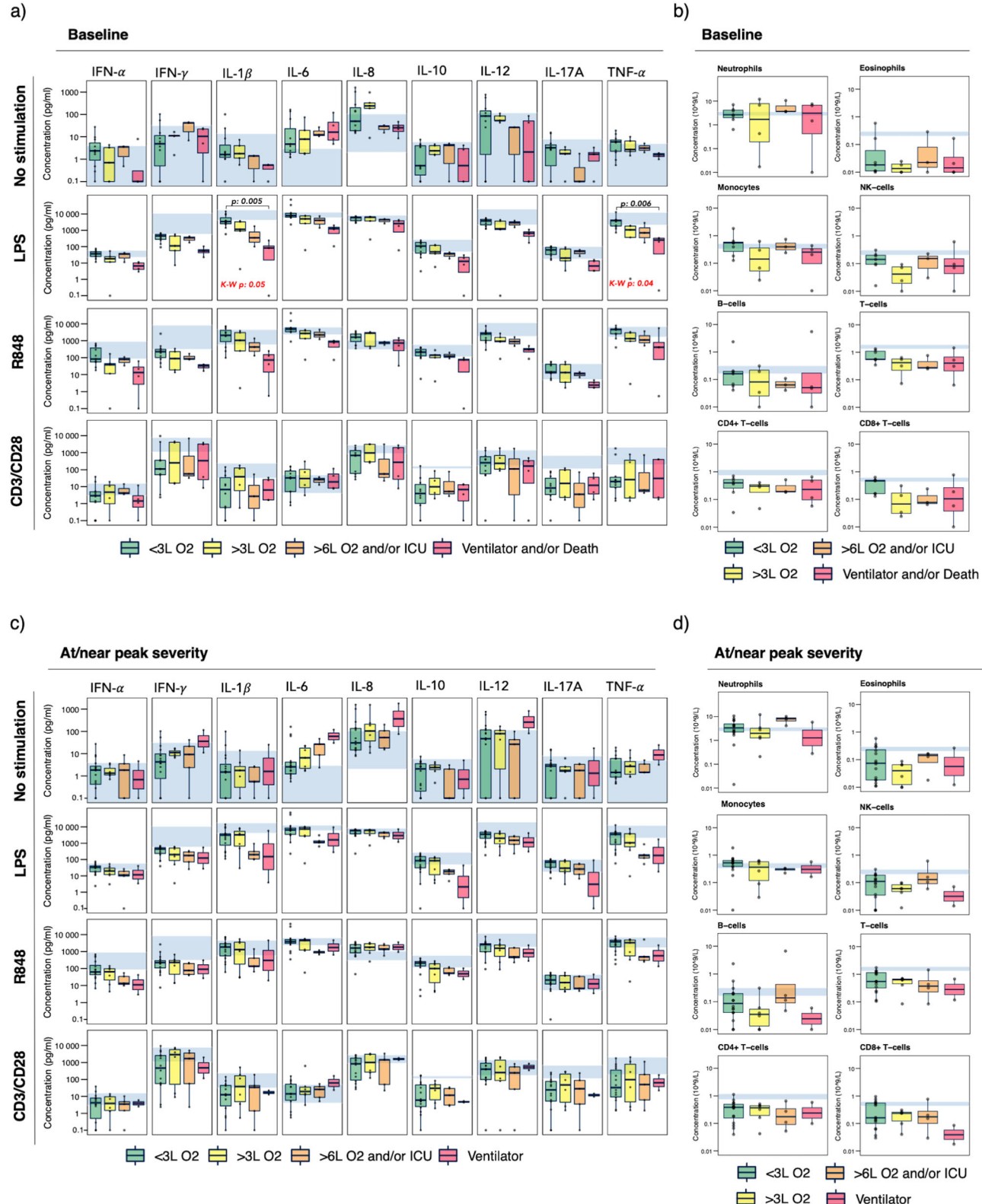

within the normal range. For LPS and R848 stimulation, we observed a trend of cytokine levels declining in a dose-response like fashion with increasing peak severity grade. This was most prominent for LPS-stimulated IL-1β and TNF-α (Fig. 2a and Supplementary Fig. 2b), both exhibiting statistically significant differences between peak severity groups (Bonferroni adjusted Dunn's post-hoc $p < 0.01$, severity group 4 vs group 1). Assessing

CD3/CD28-stimulated responses, we observed reduced levels of all cytokines except IL-6 for all four peak severity groups, but a gradual decreasing trend was not observed. The Poly I:C stimulated immune responses resembled the unstimulated cytokine values, thus not providing additional information (Supplementary Fig 3a). No statistically significant differences were observed between severity groups when assessing total immune subset cell

**Fig. 2 Stimulated immune responses and immune cell constitution at baseline vs at/near peak severity. a** Cytokine levels in response to no stimulation, LPS (bacterial), R848 (viral), and CD3/CD28 (T-cell receptor/co-receptor) at baseline ($n = 23$, except for CD3/CD28: $n = 21$). Patients are grouped based on future peak severity: Grade 1 ($n = 11$, CD3/CD28: $n = 10$, green), Grade 2 ($n = 5$, CD3CD28: $n = 4$, yellow), Grade 3 ($n = 3$, orange), Grade 4 ($n = 4$, red). **b** Immune cell subset counts at baseline ($n = 18$). Patients are grouped based on future peak severity: Grade 1 ($n = 7$, green), Grade 2 ($n = 4$, yellow), Grade 3 ($n = 3$, orange), Grade 4 ($n = 4$, red). **c** Cytokine levels in response to no stimulation, LPS, R848, and CD3/CD28 at/near peak severity ($n = 30$, except for CD3/CD28: $n = 27$). Patients are grouped based on severity grade at time of sample collection: Grade 1 ($n = 16$, CD3/CD28: $n = 14$, green), Grade 2 ($n = 7$, CD3/CD28 $n = 6$, yellow), Grade 3 ($n = 5$, orange), Grade 4 ($n = 2$, red). **d** Immune cell subset counts at/near peak severity ($n = 28$). Patients are grouped based on severity grade at time of sample collection: Grade 1 ($n = 14$, green), Grade 2 ($n = 7$, yellow), Grade 3 ($n = 5$, orange), Grade 4 ($n = 2$, red). Box edges represent the 25th and 75th percentiles, whiskers extend towards the most extreme values but no further than ± 1.5 times the interquartile range from the hinge. Hollow dots beyond whiskers represent outliers. Solid dots represent individual measurements. Blue shaded areas represent the normal reference interval. Cytokine concentration levels and immune cell subset counts are presented on a $\log_{10}$ y-axis. Severity groups were compared by the Kruskal-Wallis test and Dunn's post-hoc test, both with adjustment for multiple testing using Bonferroni. Only statistically significant adjusted p-values are shown, defined as adjusted $p < 0.05$. *LPS* lipopolysaccharide, *R848* resiquimod, *CD* cluster of differentiation, *IFN* interferon, *IL* interleukin, *TNF* tumor necrosis factor, *NK* natural killer, *L O2* liters/minute of oxygen supply, *ICU* intensive care unit.

counts in the samples where whole blood flow cytometry had been performed in parallel ($n = 18$, Fig. 2b and Supplementary Table 4). Neutrophil counts varied substantially. Monocyte counts varied without a visible pattern. Cell counts for eosinophils, natural killer (NK) cells, and T cells including CD4+ and CD8+ subsets, were suppressed compared to normal reference levels for most patients, with no difference between groups. The same was observed for B cells in all but two samples. Of note, a more in-depth characterization of the T- and B cell compartments in COVID-19 has been covered in a previous study[45]. Altogether, the declining LPS and R848 stimulated cytokine responses across clinical severity groups at baseline may thus suggest functional impairment rather than differences in immune cell constitution.

**Stimulated immune responses at clinical peak severity.** Next, we investigated stimulated immune responses as close to clinical peak severity as possible. To do so, we selected the sample closest to peak disease severity for each patient and assessed the variation of the corresponding immune responses therein ($n = 30$, Fig. 2c and Supplementary Fig. 2f–j, Supplementary Table 5). We noted a small difference between this proxy selection, defined "*at/near peak severity*", and the actual peak severity why patients were grouped based on their current severity grade at time of sample collection as illustrated in Supplementary Fig. 1b. Time from admission to peak severity is presented in Supplementary Table 13. The unstimulated levels of TNF-α, IL-6, IL-8, IL-12, and IFN-γ were elevated in patients receiving MV (severity grade 4) when compared to reference levels. Unstimulated IL-6 levels increased with increasing disease severity grade, in contrast to baseline where IL-6 levels were similar among peak severity groups (Fig. 2a). For LPS and R848 stimulated responses, a trend of declining cytokine levels with increasing severity was observed (Fig. 2c), but not as pronounced as observed at baseline. No difference was observed between severity groups for CD3/CD28-simulated cytokines responses (Fig. 2c). CD3/CD28-simulated IL-6 appeared elevated for the patients receiving MV but, considering the concentration range, this likely reflects the elevated plasma-levels observed for unstimulated IL-6. Poly I:C stimulated responses resembled the unstimulated cytokine levels. (Supplementary Fig. 3b). No statistically significant differences between severity groups were detected (Supplementary Fig. 2f–j). Immune cell subset constitution at/near clinical peak severity ($n = 28$, Fig. 2d and Supplementary Table 6) was similar to baseline, with varying neutrophil and monocyte counts as well as low eosinophil and lymphocyte counts regardless of severity grade. Overall, the unstimulated IL-6 levels, rather than stimulated cytokine responses, seemed to discriminate severity groups the best at/near clinical peak severity.

**Restored stimulated immune responses at discharge.** Next, we selected the sample closest to discharge (surviving patients only, $n = 25$, mean time from sample collection to discharge was 1,72 days, Supplementary Fig. 1c) to investigate persistence and recovery of the observed functional impairments. We defined recovery as improvement or full normalization (return to normal range) of stimulated cytokine responses. While unstimulated and Poly:IC stimulated cytokine levels tended to remain elevated at discharge, we observed an overall recovery of LPS- and R848 stimulated cytokine levels occurring within all peak severity groups (Supplementary Fig. 3c). Statistically significant improvements compared to baseline were observed for LPS-stimulated IL-1β and R848-stimulated IL-12 (adjusted $p < 0.05$ for both), and trends of improvement for LPS-stimulated IL-12 and IL-17A (adjusted $p = 0.07$ and $p = 0.09$ respectively, Fig. 3a and Supplementary Table 7). Similarly, recovery of cytokine levels was observed for CD3/CD28 stimulated responses, especially prominent for peak severity groups 2–4 (Supplementary Fig 3C), with a statistically significant improvement observed for CD3/CD28-stimulated IL-1β (adjusted $p = 0.02$, Fig. 3a). We further observed restored TruCulture immune responses at discharge regardless of previous immunosuppression (Supplementary Fig 4). Recovery was also observed within all peak severity groups for several immune cell subsets (Fig. 3b, Supplementary Fig. 3d and Supplementary Table 8), with trends towards improvement for eosinophils and CD8 + T cells (adjusted $p = 0.06$ for both).

**Loss of correlation between immune cell subsets and their cytokine response as peak severity increases.** Next, correlations between TruCulture immune responses and immune cells were investigated at baseline, at/near peak severity, and at discharge (Fig. 3c–e). At baseline and at/near peak severity, monocytes and neutrophils displayed statistically significant positive correlations with LPS-stimulated IL-8. Statistically significant correlations were also observed between monocytes and R848-stimulated IL-1β -and TNF-α at baseline, and LPS- and R848 stimulated IL-12 at/near peak severity, while neutrophils displayed statistically significant correlations with R848-stimulated IL-10 at baseline and R848 stimulated IL-8 at/near peak severity. The only statistically significant correlation between immune cell counts and cytokine response at discharge was observed between monocytes and LPS-stimulated IL-8. Generally, the majority of significant correlations were observed at baseline, and occurred between the LPS and R848-stimulated cytokine responses (Fig. 3c). Thus, the LPS and R848-stimulated cytokine responses declining with increasing peak severity, and recovery of responses observed at discharge, do not seem to be explained by changes in immune cell counts, and therefore point toward functional changes.

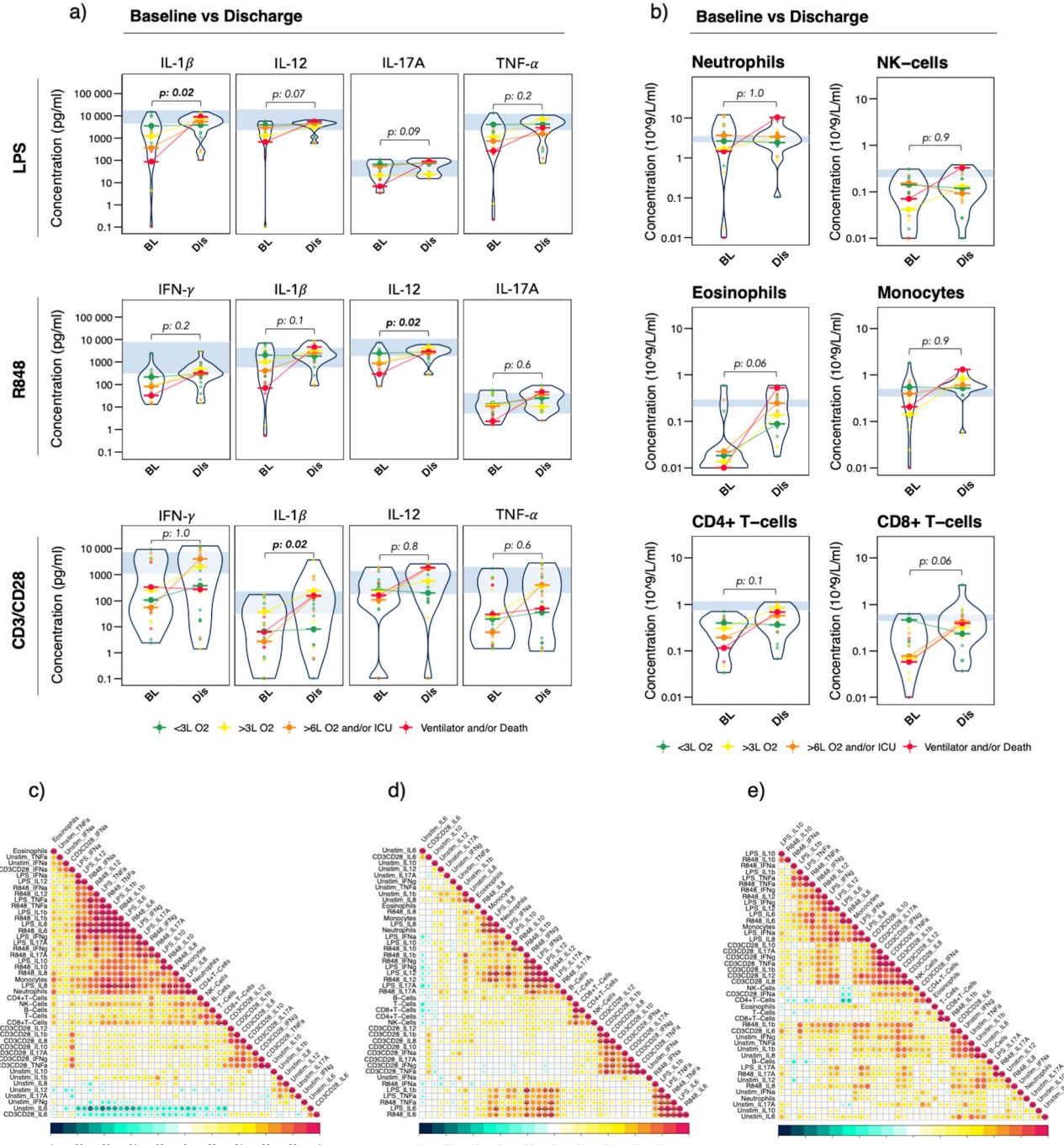

**Fig. 3 Stimulated immune responses and immune cell constitution at discharge vs baseline, and correlation between immune cell subsets and stimulated immune responses. a, b** Violin plots displaying concentration levels for (**a**) a subset of cytokines in response to LPS, R848, and CD3/CD28 at discharge ($n = 25$, except for CD3/CD28: $n = 24$) vs baseline ($n = 23$ except for CD3/CD28: $n = 21$), and **b** immune cell subsets at discharge ($n = 18$) vs baseline ($n = 28$). Solid dots represent individual patient measurements, colored by peak severity group; green = Grade 1, yellow = Grade 2, orange = Grade 3, red = Grade 4. Medians within each severity group at discharge and baseline are connected with a line, colored by peak severity. Data at baseline vs discharge were compared using the Wilcoxon signed-rank test, only including patients with paired baseline- and discharge samples available ($n = 18$ for immune responses, $n = 8$ for immune cell constitution). Adjustment for multiple testing was done using Bonferroni and adjusted $p < 0.05$ was considered statistically significant. **c–e** Correlation matrices of 8 immune cell subsets and 45 TruCulture cytokine variables by Pearson at **c** baseline, **d** at/ near peak severity, and **e** discharge. All data were log-transformed cytokine concentrations. Correlation coefficients are visualized by color intensity. Only statistically significant correlations after Bonferroni adjustment are presented;* adjusted $p < 0.05$, ** adjusted $p < 0.01$, *** adjusted $p < 0.001$. *BL* baseline, *Dis* discharge, *LPS* lipopolysaccharide, *R848* resiquimod, *CD* cluster of differentiation, *IFN* interferon, *IL* interleukin, *TNF* tumor necrosis factor, *NK* natural killer, *L O2* liters/minute of oxygen supply, *ICU* intensive care unit, *Unstim* no stimulation.

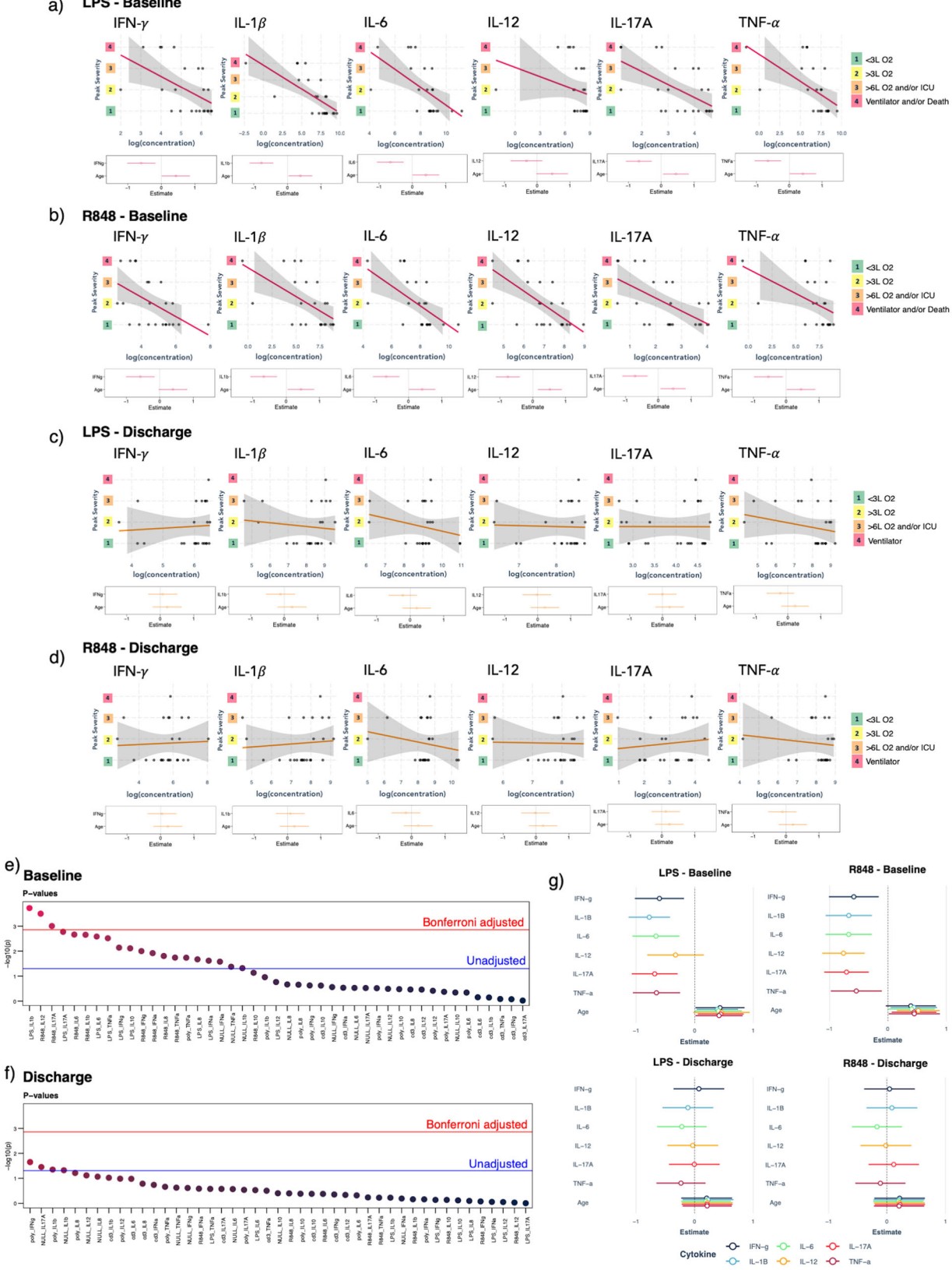

**Associations between stimulated cytokine responses and peak disease severity**. To further understand the clinical utility of this data and methodology, we investigated associations between cytokine stimulus-response (CSR) variables and peak disease severity across three clinical use-cases by ordinary least squares (OLS) (Fig. 4, Supplementary Figs. 5–7 and Supplementary

Table 9–11); (1) ability of CSR at baseline ($CSR_{baseline}$) to predict future peak severity; (2) ability of CSR at/near peak ($CSR_{peak}$) to screen current severity; and (3) ability of CSR at discharge ($CSR_{discharge}$) to infer previous peak severity. Using univariate linear models, we found no significant relationship between peak severity and clinical covariates (Supplementary Table 14). While

**Fig. 4 Associations between individual cytokine stimulus-response variables and peak severity at baseline vs discharge. a**, **b**, Associations after adjusting for age between individual cytokine variables (log-transformed cytokine concentration, log(concentration)) at baseline and future peak severity grade (Peak Severity) for **a** LPS stimulation and **b** R848 stimulation. **c**, **d** Associations after adjusting for age between individual cytokine variables (log-transformed cytokine concentration, log(concentration)) at discharge and previous peak severity grade (Peak Severity) for **c** LPS stimulation and **d** R848 stimulation. Shaded areas behind regression lines represent 95% confidence intervals. Individual regression coefficient estimates for the cytokine variable and age are illustrated in a summary plot below each regression plot, hollow dots represent the estimates, bars represent 95% confidence intervals. **e**, **f** p-values from all linear regression analyses ($n = 45$) after adjusting for age on a -$\log_{10}$-axis at **e** baseline and **f** discharge. The threshold for statistical significance is shown before adjusting for multiple tests ($p = 0.05$, blue line) and after Bonferroni-adjustment ($p = 0.001$, red line). Only associations with p-values smaller than the Bonferroni-adjusted threshold ($p < 0.001$) were considered statistically significant. **g** Regression coefficient estimates for the LPS and R848 stimulated cytokine variables and age at baseline vs recovery, hollow dots represent the estimates, bars represent 95% confidence intervals. *LPS* lipopolysaccharide, *R848* resiquimod, *CD* cluster of differentiation, *IFN* interferon, *IL* interleukin, *TNF* tumor necrosis factor, *L O2* liters/minute of oxygen supply, *ICU* intensive care unit, *NULL* no stimulation, *cd3* CD3/CD28 stimulation, *poly* Poly:IC (Polyinosinic:polycytidylic acid) stimulation.

this suggested little to no confounding in our experimental design, we decided to nevertheless adjust for age in all CSR effect size estimates given its well-established risk for severe COVID-19[16], and as it was the covariate closest to demonstrating an impact ($p = 0.08$). Given our small sample size, the other covariates were excluded to maximize statistical power. Overall, we found statistically significant linear relationships between several $CSR_{baseline}$ variables from LPS and R848 and peak severity (Fig. 4a, b, e, g; Supplementary Fig. 5). In contrast, we failed to find sufficient evidence for $CSR_{discharge}$ to infer past peak severity (Fig. 4c, d, f, g; Supplementary Fig. 6). We also observed little to no relationship between $CSR_{peak}$ and peak severity (Supplementary Fig. 7), however, the predictive signal was stronger than $CSR_{discharge}$ suggesting that collecting more samples could improve detection. These results suggest a delay between observed cytokine responses and clinical disease presentation, indicating a potential prognostic application for this method. Indeed, after correcting for multiplicity (Bonferroni, $n = 45$), statistically significant relationships between $CSR_{baseline}$ and future peak severity grade remained for LPS-stimulated IL-1β, and R848-stimulated IL-12 and IL-17A (adjusted $p = 0.008$, $p = 0.014$, and $p = 0.044$ respectively; Fig. 4a, b, e).

**A stimulation signature at baseline with potential to predict subsequent COVID-19 severity.** Having identified specific $CSR_{baseline}$ as negatively associated with subsequent disease severity, we set out to identify a combination of variables at baseline that would best predict peak severity grade. Visual inspection of standardized log-transformed cytokine concentrations shows a gradient across severity for LPS and R848 (Fig. 5a). However, correlation across LPS + R848 $CSR_{baseline}$ variables was high (Fig. 5b). Principal component analysis based on LPS + R848 data showed no clear linear relationship in eigenspace (Supplementary Fig. 8a), however, subsequent projection of data using isometric feature mapping (isomap, $k = 3$)[41] revealed a clear gradient of peak severity (Fig. 5c). We then used OLS in combination with lasso penalties to assess $CSR_{baseline}$ variables in bins of stimuli to identify the subset of predictive variables that were most likely to generalize our model while limiting potential for overfitting (Supplementary Figs. 8c–e and 9, bins are described in Methods). We focused on $CSR_{baseline}$ as the only endogenous variables to maximize power as we had already confirmed the absence of possible confounding by other exogenous factors in univariate analyses (Supplementary Table 14). Specifically, *age* was not included as its level of significance suggests its inclusion would be counterproductive. We found the best model for predicting peak severity was based on cytokine concentration data from the LPS + R848 stimuli combined ($r^2 = 0.91$, $\lambda = 0.009$, Supplementary Fig. 8c–e). The runner-up models were based on Poly:IC ($r^2 = 0.71$) and LPS ($r^2 = 0.70$) stimuli individually, while

the best model from the R848 bin explained only 40% of variance (Supplementary Fig. 8c–e). Furthermore, the LPS + R848 model performed better than the model based on only the three statistically significant variables (LPS-stimulated IL-1β, R848-stimulated IL-12, and IL-17A) from the individual OLS analyses ($r^2 = 0.48$). This suggests that the correlation between R848 and LPS responses synergistically enhances the predictive value of our model when combined. This is further supported by the relationship between R848 and LPS data projections in PCA (Supplementary Fig. 8b). At our chosen lambda, the lasso regression model for LPS + R848 excluded only four variables: LPS-stimulated IFN-γ and IL-12, and R848 stimulated IFN-α and IL-6 (Fig. 5a, b and Supplementary Fig. 8f), likely due to their high correlation with other more informative variables.

**Validation of the baseline stimulation signature in a separate cohort of hospitalized COVID-19 patients.** Next, we tested the signature identified by the LPS + R848 model in a separate cohort. We selected 20 patients with five patients in each peak severity group. The patients in the validation cohort were enrolled in the COVIMUN study in the months following the original cohort (October 2020–January 2021). Patient baseline characteristics of the validation cohort were comparable to the original cohort (Supplementary Table 15), except having a lower proportion of patients with immune dysfunction (30% vs 57%). We found that although the signature was not very predictive of specific peak severity grade vs. the rest, it performed best in predicting the two highest severity grades vs. the two lowest (sensitivity = 0.7, specificity = 0.8, diagnostic odds ratio = 9.3, Matthews correlation coefficient = 0.5, Fig. 6a and Supplementary Fig. 10). Coherently, the LPS- and R848 stimulated cytokine responses displayed the same trend of declining values with increasing peak severity grade as observed for the original cohort (Supplementary Fig. 6b). Thus, the immune signature based on LPS + R848 stimulated responses represents a model with explanatory as well as predictive value, that may improve our understanding of early functional immune impairments in COVID-19.

## Discussion
A better understanding of the immunological mechanisms driving diverse COVID-19 disease trajectories is warranted to aid the management of patients with COVID-19 and identify new targets for interventions. In this prospective study, we applied a clinically implemented and standardized analysis assessing *real-time* whole-blood stimulated immune responses by TruCulture[34,46,47] in patients hospitalized with COVID-19. We identified early functional impairments in innate immune responses at the time of hospitalization associated with subsequent COVID-19 disease

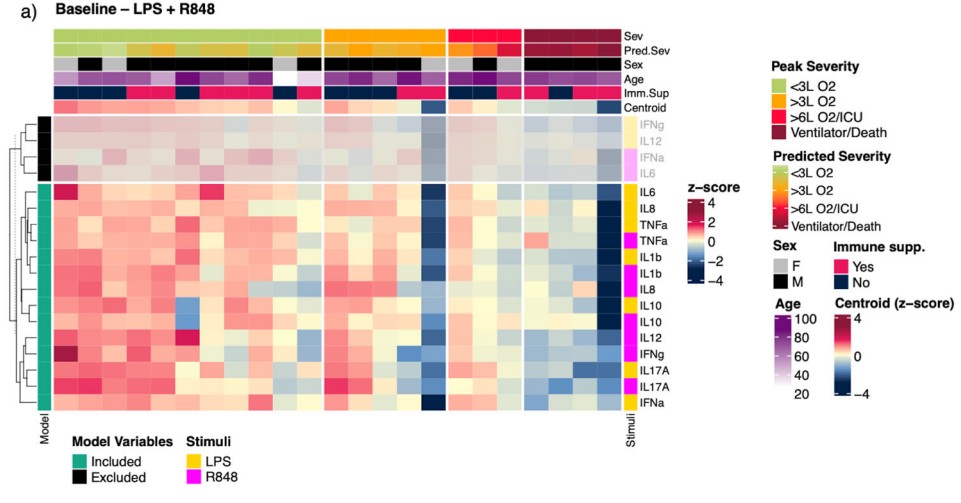

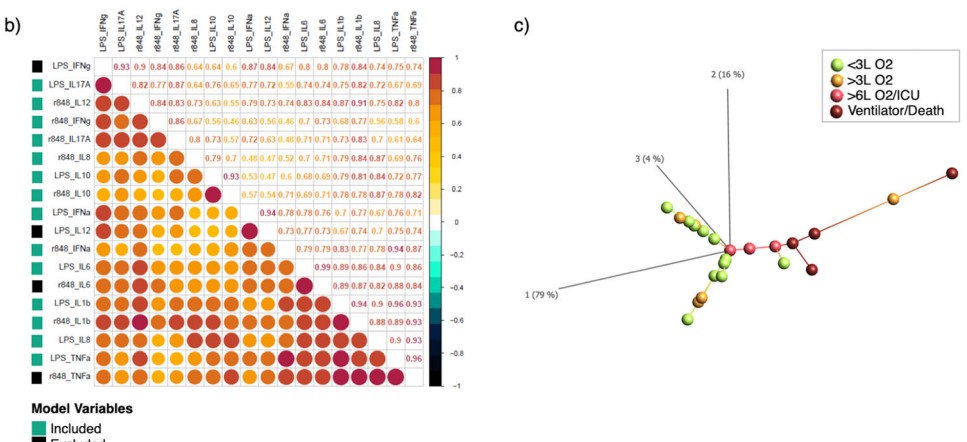

**Fig. 5 A stimulation signature based on LPS and R848 stimulated cytokine responses at baseline associated with peak severity. a** Combined LPS and R848 stimulated cytokine variables at baseline. Each column represents a patient, each row represents a stimulus-cytokine variable. Columns are grouped by future peak severity, rows are grouped by inclusion/exclusion in the LPS + R848 model. Hierarchical clustering by Euclidean distance as dissimilarity metric was preformed within groups/splits (dendrogram only shown for rows). The top annotations represent (up-down): "Sev": future peak severity, "Pred.Sev": predicted severity in current cohort based on the LPS + R848 LASSO regression model, "Sex": sex at birth, "Age": age at time of inclusion in study, "Imm.Sup": whether an immunosuppressive pre-condition was present, "Centroid": the row mean value. Row annotations represent (left-right): "Model": Inclusion/exclusion of variable in the LPS + R848 model, "Stimuli": the stimulus for each cytokine variable. Data used for visualization were log-transformed and standardized. Data from the variables excluded from the LPS + R848 model have been blurred. **b** Correlation between all LPS and R848 stimulated cytokine variables by Pearson. All data were log-transformed cytokine concentrations. Row annotation represent inclusion/exclusion of a variable in the LPS + R848 model. **c** Projection of LPS + R848 data at baseline using isomap revealing a severity gradient in the data structure. *LPS* lipopolysaccharide, *R848* resiquimod, *L O2* liters/minute of oxygen supply, *ICU* intensive care unit.

severity, and observed reconstitution of these impairments in recovering patients regardless of previous disease severity. Furthermore, we identified individual immune response biomarkers associated with subsequent disease severity, and illustrate that LPS- and R848 stimulated responses combined exhibit the potential to constitute a predictive signature for the identification of patients with high risk of severe COVID-19 to be further optimized.

Previous studies have highlighted elevated circulating levels of inflammatory cytokines in early disease to be associated with developing severe COVID-19[22–24]. Here, unstimulated cytokine levels were close to or within the normal range at baseline for all severity groups, while elevated levels were first observed at/near peak severity for patients with severity grade 4. This discrepancy may reflect different timing for sample collection, where the elevated baseline cytokine levels demonstrated in previous studies

may represent patients infected earlier or with progressive disease already at the time of hospitalization[22,24]. However, due to differences in measurement techniques- and conditions between TruCulture and other assays determining plasma/serum cytokine levels, unstimulated cytokine concentrations presented here may not be directly comparable to plasma/serum cytokine concentrations presented in other studies.

Intriguingly, we observed distinct trends of LPS- and R848 stimulated cytokine responses at baseline declining with increasing grade of subsequent peak disease severity, and identified LPS-stimulated IL-1B and IL-17A, and R848-stimulated IL-12, as individual baseline biomarkers significantly associated with subsequent peak severity. However, only a few LPS and R848-stimulated cytokine variables, such as IL-8, were significantly correlated with immune cell subset counts at baseline. Importantly, we did not observe a significant correlation between any of

a)

| | Grade 1 | Grade 1-2 | Grade 3-4 | Grade 4 |
|---|---|---|---|---|
| Sensitivity: | 0,6 | 0,8 | 0,7 | 0 |
| Specificity: | 0,67 | 0,7 | 0,8 | 0,87 |
| PPV: | 0,375 | 0,73 | 0,78 | 0 |
| NPV: | 0,83 | 0,78 | 0,73 | 0,72 |
| FPR: | 0,33 | 0,3 | 0,2 | 0,13 |
| FNR: | 0,4 | 0,2 | 0,3 | 1 |
| FDR: | 0,625 | 0,27 | 0,22 | 1 |
| FOR: | 0,17 | 0,22 | 0,27 | 0,28 |
| DOR: | 3 | 9,33 | 9,33 | 0 |
| MCC: | 0,24 | 0,5 | 0,5 | -0,19 |

b)

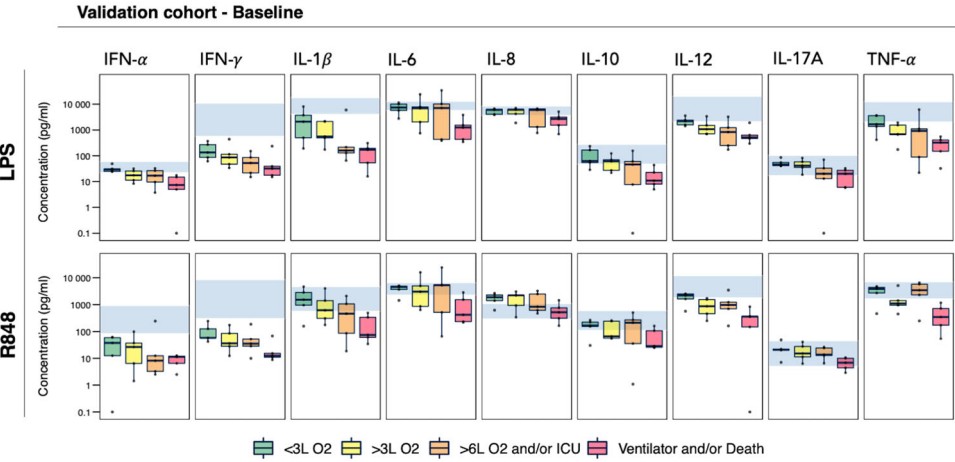

**Fig. 6 Validation of the immune response signature in a separate cohort.** The LPS + R848 model was validated on a separate cohort of hospitalized COVID-19 patients (n = 20, 5 in each peak severity group). **a** Sensitivity/recall, specificity, false positive rate (FPR), false negative rate (FNR), positive predictive value (PPV) /precision, negative predictive value (NPV), false discovery rate (FDR), false omission rate (FOR), diagnostic odds ratio (DOR), and Mathew's correlation coefficient (MCC) are presented for predicting severity grade 1 alone, grade 1–2, grade 3–4, and grade 4 alone. **b** Cytokine levels in response to LPS and R848 based on baseline data from the validation cohort (n = 20). Patients are grouped based on future peak severity: Grade 1 (n = 5, green), Grade 2 (n = 5, yellow), Grade 3 (n = 5, orange), Grade 4 (n = 5, red). Box edges represent the 25th and 75th percentiles, and whiskers extend towards the most extreme values but no further than ± 1.5 times the interquartile range from the hinge. Hollow dots beyond whiskers represent outliers. Solid dots represent individual measurements. Blue shaded areas represent the normal reference interval. Cytokine concentration levels and immune cell subset counts are presented on a log10 y-axis. *LPS* lipopolysaccharide, *R848* resiquimod, *IFN* interferon, *IL* interleukin, *TNF* tumor necrosis factor, *L O2* liters/minute of oxygen supply, *ICU* intensive care unit.

the LPS and R848-stimulated cytokine variables that displayed significant associations with peak severity and immune cell subset counts including monocytes, which represent an acknowledged source of LPS- and R848 stimulated cytokines[48,49]. Thus, changes in immune cell counts cannot alone account for the pattern observed for LPS- and R848 stimulated cytokine levels. Notably, the few patients who had received dexamethasone, as well as patients with pre-existing immunosuppressive conditions and/or comorbidities, were evenly distributed across all severity groups, indicating that these factors were not the main contributor to the findings in this study. Suppressed expression of cytokines like TNF-α by innate immune cells in response to viral and bacterial TLR stimulation in COVID-19 patients compared with healthy individuals has previously been reported[25]. These findings were based on samples collected during ongoing disease, thus furthering the findings of this study. Thus, this is to our knowledge

the first study demonstrating early impaired innate immune responses based on TLR-stimulation of fresh whole-blood samples collected from COVID-19 patients at the time of hospitalization, that furthermore shows an association with subsequently developed disease severity. Importantly, most patients had severity grade 1 at the time of baseline sampling (Supplementary Fig. 1a), indicating that our findings expose early functional immune impairments preceding both disease progression and increased systemic cytokine levels.

Interestingly, we observed improved innate- and T cell stimulated responses in surviving patients at discharge, regardless of previous disease severity or immunosuppressive pre-condition. This was paralleled by restored immune cell levels, especially of CD8 + T cells, in coherence with previous studies highlighting the importance of adaptive T cell-mediated responses for the clearance of SARS-CoV-2 infection[45,50,51]. Such synchronized

reconstitution of innate and adaptive responses, at both a functional and cellular level, associated with recovery regardless of previous disease severity, has to our knowledge not previously been demonstrated in a longitudinal study of COVID-19.

When conducting individual OLS analyses, variables expected to affect outcome were investigated in univariate linear regressions, but found to have no impact in this study, likely due to the small sample size. Thus, to maximize power, age was the only covariate included in the analyses, where the strongest associations between individual cytokine responses and peak severity were stimulated by LPS and R848 at baseline. In coherence, lasso regression based on combined LPS- and R848 data at baseline provided the best model for predicting subsequent peak severity in our cohort. To our surprise, the model based on Poly:IC-data alone provided one of the highest $r^2$ among the individual stimuli ($r^2 = 0.71$), in contrast to what we had expected based on the OLS with Poly:IC CSR variables (Supplementary Fig. 5e–f). Interestingly, out of four tested combinations of individual stimuli, LPS + R848 was the only combination that improved $r^2$ compared to the individual stimuli models, also providing the highest $r^2$ of all models. Adding age to the bins had little to no impact, reduced $r^2$ for the best performing models, and improved $r^2$ for poor performing models. Given that age is a well-established risk factor for severe COVID-19, this sensitivity analysis indicates that it likely serves as a proxy for more precise signals represented by LPS- and R848 stimulated immune responses in our patient cohort. However, this warrants further investigation in larger cohorts. Our findings further imply that the LPS-and R848 stimulated immune responses may reflect two distinct, but complementary, immune pathways where potential impairments in either may impact the other, as further supported by the high correlation observed between variables (Fig. 5b).

Importantly, the LPS + R848 model was superior to the model based on only the three significant variables identified in the individual OLS analyses. This supports that the lack of additional significant linear relationships between individual CSR variables and severity was likely a power issue. Given the small cohort size ($n = 23$), large number of predictors (18), and high $r^2$ (0.91), we expected the LPS + R848 model to be overfitted to this specific cohort. This was in part reflected in the lower accuracy of the model to predict one specific peak severity grade vs the rest when tested on the validation cohort. The presence of two pronounced outliers, uneven distribution of patients in peak severity groups in the original cohort, along with a larger proportion of subjects with immune dysfunction in the original cohort, likely contributed further to the performance. Nevertheless, when reducing the predicted outcome to two instead of four severity grades (grade 3–4 vs grade 1–2), the model identified 7 out of 10 patients with risk of needing >6 L of O2, admission to ICU, need for MV, or death. From a clinical perspective, this could provide valuable information allowing for closer monitoring and earlier interventions. While supporting the predictive value of the LPS- and R848-based immune signature, the results from the validation cohort highlight that further development and optimization of this model is warranted. Exploring the signature in a larger cohort, adjusting the model output limits defining a specific peak severity grade, and investigating threshold levels of cytokine concentrations constitute a few parameters that could be optimized to improve the accuracy and performance.

While its role as a predictive clinical tool warrants further study, we propose that the LPS + R848 stimulated immune signature represents a valuable explanatory model indicating an important role of these immune pathways (activating TLRs 4 and 7/8 respectively) in early COVID-19 pathogenesis. Induction of type I IFNs and pro-inflammatory cytokines through activation of TLRs constitutes a key mechanism of the initial innate host defense against infectious threats[52]. With LPS being a TLR4 agonist, and R848 being an agonist of TLR7/8[53,54], our findings point toward early impairments in such key innate immune activation cascades. Notably, SARS-CoV-2 has been shown to activate both TLR7/8[55], and TLR4[56], therefore both the R848 and LPS stimulated responses may mimic the in vivo anti-COVID-19 responses. In line with previous studies demonstrating impaired type I IFN-responses being associated with developing severe COVID-19[25,26], LPS-stimulated IFN-α was included in the LPS + R848 model, indicating lower levels a potential predictor for severe COVID-19. SARS-CoV-2 has been shown to interfere with the crucial early type I IFN-response, permitting further replication and tissue damage[57]. This may delay the infiltration of inflammatory cells into infected tissues, which may further on elicit an imbalanced inflammatory exacerbation driving the progression of severe disease[58]. Assembly of the inflammasome with subsequent induction of IL-1β and further downstream IL-6, constitutes a key downstream effector mechanism of type I IFNs in activating an acute inflammatory anti-viral response[59]. LPS-stimulated IL-1β displayed the strongest negative association with peak severity, and stimulated IL-1β (LPS and R848) and IL-6 (LPS) were included in the LPS + R848 model. This may imply a lack of adequate inflammasome activity in the early defense against SARS-CoV-2[60–62]. IL-12 derived from innate cells including monocytes and dendritic cells constitute a crucial component in the activation and polarization of IFN-γ producing type 1 T helper (Th1) cells[63], which characterize the SARS-CoV-2 specific CD4+ T cell population previously described[50,51]. Importantly, R848-stimulated IL-12 and IFN-γ were included in the LPS + R848 model, and R848-stimulated IL-12 alone exhibited a significant negative association with COVID-19 severity, indicating early defects in this crucial link between innate and adaptive immune responses significantly impacting disease severity. The importance of this IL-12 mediated link is being exploited to improve vaccination responses in a current clinical trial of a DNA-vaccine containing an added IL-12-plasmid (NCT04627675). Thus, the LPS- and R848 stimulated TruCulture responses likely capture several steps of an early immune cascade that lacks appropriate activation, possibly due to initial impairments in the induction of an adequate type I IFN response.

Here we describe the implementation of the standardized TruCulture assay, where the immune responses stimulated by LPS and R848, specifically, enabled identification of patients harboring transient impairments in early immune response at time of hospitalization with COVID-19 infection. We identified an immune response signature based on LPS + R848 stimulated responses providing insights into early COVID-19 pathology as well as carrying potential to identify patients at risk of developing severe COVID-19. While investigations to further explore and validate our findings are currently ongoing at our institution, we also urge other researchers to pursue assessment of functional impairments in COVID-19. Such continued joined efforts could ultimately enable the development of a validated tool for the early identification of high-risk patients with COVID-19 to be tested in early intervention studies as well as to aid identification of potential targets for immune modulation in COVID-19.

## Data availability

This work was based on a small cohort of subjects who are part of the bigger COVIMUN study. Raw data cannot yet be made publicly available due to data privacy issues according to EU legislation. However, anonymized source data for each figure is provided in Supplementary Datas 1 and 2. Raw data will be made available in accordance with EU regulations for data privacy once these have been published for the entire cohort— hopefully within 12 months, please contact the corresponding author regarding requests.

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

## Acknowledgements

The authors would like to acknowledge all the investigators and patients participating in this study. This work was supported by a grant from the Ministry of Higher Education and Science, Copenhagen, Denmark (0238-00006B) and was further supported by funding from the Lundbeck Foundation (R349-2020-835, received by ZBH).

## Author contributions

S.R.O. and C.N. designed and initiated the study and share senior authorship. R.S., A.H., and S.D.N. collected the data, and R.S. analyzed the data and wrote the first draft of the manuscript. C.M. contributed with supervision and input for data analysis, discussion, and interpretation of results. A.Z., R.A., T.F., and L.K.G. gave input to data visualization, analysis, and interpretation. L.K.G., M.A.A., C.C.B., C.B., C.N., and S.R.O. contributed to the discussion and clinical interpretation of data and results. S.R.O. was responsible for the TruCulture analysis, and provided the TruCulture data. J.B. and H.M. analyzed and provided the flow cytometry data, and contributed to data interpretation. P.T.B., B.L., A.S., A.O.G., D.S.H., M.M., D.P., Z.B.H., and C.B. helped to include patients in the study and collect clinical data. J.L., A.M.L., and M.H. contributed to the management of the study and to the discussion of the results. All authors participated in writing and editing the final manuscript.

## Competing interests

The authors declare no competing interests.

## Additional information

[1]Department of Hematology, Copenhagen University Hospital, Rigshospitalet, Copenhagen, Denmark. [2]PERSIMUNE Center of Excellence, Copenhagen University Hospital, Rigshospitalet, Copenhagen, Denmark. [3]Department of Clinical Immunology, Copenhagen University Hospital, Rigshospitalet, Copenhagen, Denmark. [4]Department of Pulmonary and Infectious Diseases, Copenhagen University Hospital, Nordsjællands Hospital, Hillerød, Denmark. [5]Department of Clinical Medicine, University of Copenhagen, Copenhagen, Denmark. [6]Department of Virus & Microbiological Special Diagnostics, Division of Infectious Disease Preparedness and Research, Statens Serum Institut, Copenhagen, Denmark. [7]Department of Medicine, Section of Infectious Diseases, Herlev and Gentofte Hospital, Herlev, Denmark. [8]Department of Infectious Diseases, Copenhagen University Hospital, Rigshospitalet, Copenhagen, Denmark. [9]Department of Respiratory Medicine, Bispebjerg Hospital, Copenhagen, Denmark. [10]These authors jointly supervised this work: Carsten Utoft Niemann, Sisse Rye Ostrowski. ✉email: sisse.rye.ostrowski@regionh.dk

