## [Peer Review File · Communications Medicine]

Reviewers' comments:

Reviewer #1 (Remarks to the Author):

In the present manuscript, Svanberg and colleagues analyze the blood of 30 hospitalized COVID-19 patients by using TruCulture, an assay that measures selected cytokines in unstimulated cells or upon PRR or TCR stimulation. The authors test these responses longitudinally on hospitalized SARS-CoV-2 positive patients at time of admission (baseline), peak of disease severity and time of discharge. They also assess immune cell counts in the blood. They identify the combination of LPS- and R848-induced responses at baseline as the most predictive of disease outcome. These findings suggest that impairment of the early innate immune response to SARS-CoV-2 contributes to severe disease development. Moreover, if validated, this strategy could be adopted for early identification of high-risk patients upon hospital admission.

Although potentially interesting, the extremely limited number of patients present in each of the 4 severity categories utilized by the author is a major concern. Similarly, although statistical analyses are utilized in Figures 4 and 5, no statistic is reported for figure 2 and 3. Without an appropriate statistical analysis, no conclusions can be driven for figure 2 and 3. The authors must provide this analysis and if no significance is reached for the parameters analyzed and discussed, more subjects need to be recruited to the study. In particular:

- 1) In Figure 2 and Figure 3 and Supplementary Figure 2 no statistical analysis supports the observed trends in cytokine levels/cell counts increasing or decreasing with different peak severity grades compared to the reference interval.
- 2) In Figure 2 and Figure 3 and Supplementary Figure 2 comparisons between different severity groups could be performed.
- 3) In Figure 3 and Supplementary Figure 2 comparisons between baseline and discharge values could be performed.
- 4) In Figure 2 and 3, a statistical correlation between the immune populations present in each sample and the levels of cytokines should be performed. It is key to determine if levels of specific cytokines are linked to decreased or increased level of specific immune cells.

Other points:

- 1) Lines 111-114: please acknowledge that regulation of IFN production is more complex, based on findings from Zhou Z. et al, 2020, Cell Host and Microbe; Lee JS. et al Sci. immunology 2020; Lucas C. et al Nature, 2020; Ziegler CGK et al, Cell 2021; Sposito B. et al Cell, 2021.
- 2) It would be important to grade COVID-19 severity based on WHO parameters rather than with the 4 classes identified here. At least, describe the "conversion" to WHO severity as identified by WHO guidelines.
- 3) How is the normal reference level for each parameter analyzed in figure 2 determined?
- 4) For figure 2C, it would be important to know the number of days each subjects took to get to disease peak.
- 5) Extended Data Figure 4, 5, 6 are not cited correctly in "Results" section.
- 6) Figure 4 and Extended Data Figure 4, 5, 6. Clarify how the variable "Estimate" is calculated.

Reviewer #2 (Remarks to the Author):

Svanberg et al. analyzed immune responses in 30 COVID-19 patients of varying disease severity using Trucount immune assay and flow cytometry. Their objective was to determine an early correlate that predicts disease severity. The study was well-designed and performed carefully; however, suffers from a lack of novelty. Several studies have shown impaired innate immune response, peripheral immune suppression despite enhanced cytokine response as a signature of severe COVID-19. The authors cite some but not all (for instance, Matthew et al. Science 2020) relevant studies. While the introduction has covered the most relevant details, the authors haven't cited appropriate literature. For e.g., Arunachalam et al. 2020 showed almost all that's described in lines 109-117 and the findings of the current study but has been cited somewhere in the discussion.

Fig.2-3: Are any of the differences statistically significant? Please provide statistical analyses.

The argument that these signatures could be identified at the time of hospital admission, and therefore is predictive of the disease trajectory is valuable, but such a claim needs validation. Could the authors test their signature in an additional set of patients?

Reviewer #3 (Remarks to the Author):

This article shows that the potential to predict the outcome of the COVID-19 severity using whole blood samples at the early stage of the course of symptom (at the time of hospital admission). The results and conclusions from the study show the novel information that helps decision making for those who work at clinical sites. The overall manuscript is well written, and the methodologies (including statistical analysis) were properly performed. Two statistical models were built for clinical data showing a similar trend result, which strengthen the hypothesis of this study. This study analysed pilot data with small samples, but the results obtained from this study contains crucial evidence to encourage further study in the future applying wider data. Several points that improve the validity of this study and understanding for readers, which I believe, are the following:

1. There are several approaches to select covariates for prediction models. The methods applied here [based on statistical significance for the OLS model (age) and based on a priori knowledge for potential confounders for the OLS with lasso model (6 variables)] are widely used approaches. However, why the inconsistent selection criteria were used for both methods was not explained. Were the criteria of covariates selection designed before the analysis? Is the reason for selecting one covariate (i.e., age) for the OLS model also due to the small sample size? The different trend between the two models shown in this study (lines 329–331) might be influenced by these confounder adjustments. Please consider explaining why the different selection criteria were used for the models in the method section and discuss potential/expected bias from the adjustments.
2. The same as the above issue. Lines 329–331: this could be (could be not though) because of not controlling confounders other than age. Please discuss the limitation if it is difficult to adjust confounders (e.g. due to small sample size or other reasons).
3. Line 234, Fig.3D: there is no Fig. 3D panel. Is it Fig.3A?
4. Lines 278-280, "This suggests that despite ... when combined.": can you show the evidence of this significant orthogonal effect from the analysis not only from the difference of r^2 ? If yes, please add the analysis. If not, please move this interpretation to the discussion section only (together with

lines 331–336).

5. Lines 312–316: how evenly the patients (with dexamethasone-treatment/immunosuppressive-conditions/commodities) were distributed was not explained. Proportionally even or the absolute numbers of them were even among groups? It is good for readers to show the proportion of each condition.

6. Lines 452–460: do authors consider that not all missingness were at random (just some of them were mentioned as missing at random)? How they were missing and how this missingness influenced the result should be discussed.

7. Lines 522–524: the principal component analysis did not show the same trend as Isomap analysis, but it is good to show the figure of PCA result (the similar type of the plot of Isomap analysis) in the supplementary material.

8. Lines 541–543: does “ r^2 ” stand for the “coefficient of determination”? Please define in the method part. Is the equation (line 543) properly shown (or typo)? It seems that the denominator and numerator are opposite. Also, since the predictions were made for individual-level, the notation of the summation should be used.

Response to Reviewers' Comments (author comments in blue):

To the Reviewers,

We are very grateful for the many relevant comments provided by the three reviewers.

We have complied with as many as possible of these, and believe that they have considerably improved the manuscript, which we hope will be suitable for publication in its current version.

For simplicity, the reviewer comments/questions are in bold and our response is in blue and red (the latter when based on novel data included in the manuscript).

Best Regards,

Rebecka on behalf of all authors

Before point-by-point responses to each reviewer we would like to make all three reviewers aware of a couple of updates made in the introduction. These changes have been made due to progressions of the pandemic and changes in its management that have occurred since the manuscript was initially submitted, thereby ensuring that the introduction is accurate and up-to-date.

Changes in manuscript:

Line 91-105: The coronavirus disease 2019 (COVID-19) pandemic caused by severe acute respiratory syndrome coronavirus 2 (SARS-CoV-2) remains a global health crisis, having already claimed **over 6 million lives by January 2022**.^{1,2} The clinical presentation and disease course of COVID-19 is heterogenous, varying from asymptomatic or mild symptoms to severe pneumonia with acute respiratory distress syndrome (ARDS) requiring mechanical ventilation (MV) or septic shock with multi-organ failure.^{3,4} Severe symptoms usually develop within 1-2 weeks after symptom onset.^{5,6} During the first pandemic waves, approximately 15% of SARS-CoV-2 PCR-positive cases developed severe disease, and 5% required intensive care and/or MV.⁷⁻¹⁰ **While the emergence of vaccines has remarkably improved these outcomes,¹¹⁻¹³ hospitalization due to COVID-19 still entails risk for critical disease and death, especially among patients who are unvaccinated or have insufficient or declining vaccine response.^{14,15} Risk factors for severe disease and death among both unvaccinated and vaccinated patients include older age, male gender, and pre-existing comorbidities such as obesity, hypertension, and type 2 diabetes, as well as conditions associated with immunosuppression.^{16,17} Despite improvements in disease-related outcomes, COVID-19 still challenges health care systems worldwide, warranting means for upfront risk stratification of patients at time of admission.**

Reviewer #1 (Remarks to the Author):

In the present manuscript, Svanberg and colleagues analyze the blood of 30 hospitalized COVID-19 patients by using TruCulture, an assay that measures selected cytokines in unstimulated cells or upon PRR or TCR stimulation. The authors test these responses longitudinally on hospitalized SARS-CoV-2 positive patients at time of admission (baseline), peak of disease severity and time of discharge. They also assess immune cell counts in the blood. They identify the combination of LPS- and R848-induced responses at baseline as the most predictive of disease outcome. These findings suggest that impairment of the early innate immune response to SARS-CoV-2 contributes to severe disease development. Moreover, if validated, this strategy could be adopted for early identification of high-risk patients upon hospital admission.

Although potentially interesting, the extremely limited number of patients present in each of the 4 severity categories utilized by the author is a major concern. Similarly, although statistical analyses are utilized in Figures 4 and 5, no statistic is reported for figure 2 and 3. Without an appropriate statistical analysis, no conclusions can be driven for figure 2 and 3. The authors must provide this analysis and if no significance is reached for the parameters analyzed and discussed, more subjects

need to be recruited to the study. In particular:

1) In Figure 2 and Figure 3 and Supplementary Figure 2 no statistical analysis supports the observed trends in cytokine levels/cell counts increasing or decreasing with different peak severity grades compared to the reference interval.

A: We are aware of, and agree with, the fact that the number of subjects is a limitation to this study. The visual trend observed in Fig 2, 3, and Supplementary Fig 2, that cytokine levels/cell counts at baseline decrease with increasing severity grade is not supported statistically in these figures, however we proceeded to test the hypothesis by assessing associations between cytokine levels/cell counts and severity grades in the OLS analyses (Fig 4, new Supplementary Fig 5-6). Therefore we chose to apply different statistical analyses to assess differences between severity groups in Fig 2, 3 and Supplementary Fig 2 as addressed under point 2.

2) In Figure 2 and Figure 3 and Supplementary Figure 2 comparisons between different severity groups could be performed.

A: Thank you for the suggestion to add further statistical analyses to support the visual trends observed in Fig 2, 3 and Supplementary Fig 2, we believe that this has significantly improved the paper.

To address the raised issue we have used Kruskal-Wallis test (due to the data not displaying a normal distribution) to assess differences across severity groups. We adjusted for multiple comparisons using Bonferroni within each stimulus for TruCulture data (9 cytokines = 9 tests) and within the total number of immune cell populations (8 populations = 8 tests). The Kruskal-Wallis test was statistically significant (adjusted $p < 0.05$) for LPS-stimulated IL-1 β and TNF- α at baseline, so for these variables we applied Dunn's post-hoc test (with Bonferroni method). We believe that applying these analyses further supports our thesis, as we demonstrate two statistically significant tests when applying a conservative method to adjust for multiplicity despite the limited number of subjects. While the lack of additional statistically significant tests is likely an issue of power, our observation of cytokine concentrations declining with peak severity grade at baseline provided the foundation to investigate these trends through OLS and LASSO regression, as demonstrated in the rest of the paper. Therefore, we did not consider it necessary to add more subjects to the study.

Figures have been updated with adjusted p-values for the statistically significant tests, and we have added a supplementary figure (new Supplementary Fig 2) with the sensitivity analyses from the Kruskal-Wallis tests at baseline and at/near peak severity.

Changes in manuscript:

Line 188-191: For LPS and R848 stimulation, we observed a visual trend of cytokine levels declining in a dose-response like fashion with increasing peak severity grade. This was most prominent for LPS-stimulated IL-1 β and TNF- α (Fig 2A, Supplementary Fig 2B), both exhibiting statistically significant differences between peak severity groups (Bonferroni adjusted Dunn's post-hoc $p < 0.01$, severity group 4 vs group 1).

Line 194-196: No statistically significant differences were observed between severity groups when assessing total immune subset cell counts in the samples where whole blood flow cytometry had been performed in parallel ($n=18$, Fig 2B).

Line 220-221: No statistically significant differences between severity groups were detected (Supplementary Fig 2F-J).

Online methods, line 565-573: Distribution of TruCulture cytokine concentrations and DuraClone immune cell subset counts between peak severity groups at baseline and at/near peak severity were compared using the Kruskal-Wallis test with adjustment for multiple testing using Bonferroni (9 tests for TruCulture data; 9 cytokines within one stimulus, 8 tests for DuraClone data; 8 immune cell subsets). Post-hoc Dunn's test was performed where the Kruskal-Wallis

test was statistically significant after adjustment for multiplicity, also using Bonferroni to adjust for multiple testing (adjustment for 6 tests; Grade 1 vs 2, Grade 1 vs 3, Grade 1 vs 4, Grade 2 vs 3, Grade 2 vs 4, and Grade 3 vs 4). Adjusted $p < 0.05$ was considered statistically significant. Summary statistics from all Kruskal-Wallis tests at baseline and at/near peak severity are presented in Supplementary Tables 10-13.

See new Supplementary Fig 2 and Supplementary tables 10-13.

Supplementary figure legend 2, line 78-84: **Supplementary Fig. 2. Significance of tested comparisons between severity groups at baseline and at/near peak severity** . a-e, p-values from Kruskal-Wallis tests comparing severity groups for each stimulus-response variable at baseline. f-j, p-values from Kruskal-Wallis tests comparing severity groups for each stimulus-response variable at/near peak severity. P-values are presented on a $-\log_{10}$ -axis. The threshold for statistical significance is shown before adjusting for multiple tests ($p = 0.05$, blue line) and after Bonferroni-adjustment (9 tests: $p = 0.006$, red line). Only p-values smaller than the Bonferroni corrected threshold were considered statistically significant.

Supplementary table legend 10, line 38-41: **Supplementary table 10. Summary statistics from Kruskal-Wallis tests comparing cytokine concentrations at baseline**. Summary statistics for Kruskal-Wallis tests comparing distribution of TruCulture cytokine concentrations between peak severity groups at baseline. Bonferroni adjustment is performed within each stimulus (9 tests).

Supplementary table legend 11, line 43-46: **Supplementary table 11. Summary statistics from Kruskal-Wallis tests comparing cytokine concentrations at/near peak severity**. Summary statistics for Kruskal-Wallis tests comparing distribution of TruCulture cytokine concentrations between peak severity groups at/near peak severity. Bonferroni adjustment is performed within each stimulus (9 tests).

Supplementary table legend 12, line 48-51: **Supplementary table 12. Summary statistics from Kruskal-Wallis tests comparing immune cell subset counts at baseline**. Summary statistics for Kruskal-Wallis tests comparing distribution of DuraClone immune cell subset counts between peak severity groups at baseline. Bonferroni adjustment is performed for 8 tests.

Supplementary table legend 13, line 53-56: **Supplementary table 13. Summary statistics from Kruskal-Wallis tests comparing immune cell subset counts at/near peak severity**. Summary statistics for Kruskal-Wallis tests comparing distribution of DuraClone immune cell subset counts between peak severity groups at/near peak severity. Bonferroni adjustment is performed for 8 tests.

3) In Figure 3 and Supplementary Figure 2 comparisons between baseline and discharge values could be performed.

A: Thank you for the suggestion to apply statistical analyses to support the observation of cytokine concentrations and immune cell numbers improving or normalizing at discharge compared to baseline. Due to the small number of subject in each group it was not feasible to perform paired analyses assessing discharge vs baseline within severity groups. However, since our observation and main point was that improvement/normalization occurred regardless of previous severity grade, we analyzed the groups pooled together. We applied paired Wilcoxon signed-rank tests, including only the subjects with paired samples available at discharge and baseline. Adjustment for multiplicity was again performed using Bonferroni within each stimulus for TruCulture data (9 cytokines = 9 tests) and within the total number of immune cell populations (8 populations = 8 tests). After adjustment, we found statistically significant improvements for LPS-stimulated IL-1B, R848-stimulated IL-12, and CD3CD28-stimulated IL-1B, while additional cytokines as well as immune cell subsets displayed near-significant trends.

We display the results of these new analyses in a new Fig 3A-B, with violin plots illustrating the change in distribution of the data at discharge vs baseline (an overall wider distribution at baseline), and additionally we display the medians of the severity groups, and mark the change of these medians from baseline to discharge with a connecting line. This illustrates that for most cytokines and immune cell subsets, improvement occurs within all four severity groups. The old Fig 3 is moved to Supplementary Fig 3C-D.

Changes in manuscript:

Line 234-242: Statistically significant improvements compared to baseline were observed for LPS-stimulated IL-1 β and R848-stimulated IL-12 (adjusted $p < 0.05$ for both), and trends of improvement for LPS-stimulated IL-12 and IL-17A (adjusted $p = 0.07$ and $p = 0.09$ respectively, Fig 3A). Similarly, recovery of cytokine levels was observed for CD3/CD28 stimulated responses, especially prominent for peak severity groups 2-4 (Supplementary Fig 3C), with a statistically significant improvement observed for CD3/CD28-stimulated IL-1 β (adjusted $p = 0.02$, Fig 3A). We further observed restored TruCulture immune responses at discharge regardless of previous immunosuppression (Supplementary Fig 4). Recovery was also observed within all peak severity groups for several immune cell subsets (Fig 3B, Supplementary Fig 3D), with trends towards improvement for eosinophils and CD8+ T cells (adjusted $p = 0.06$ for both).

Online methods, line 573-579: TruCulture cytokine concentrations and DuraClone immune cell subset counts at baseline vs discharge were compared using the Wilcoxon signed-rank test, only including patients with paired baseline- and discharge samples available ($n = 18$ for TruCulture, $n = 8$ for DuraClone). Adjustment for multiple testing was done using Bonferroni (9 tests for TruCulture data; 9 cytokines within one stimulus, 8 tests for DuraClone data; 8 immune cell subsets), and adjusted $p < 0.05$ was considered statistically significant. Summary statistics from all Wilcoxon signed-rank tests are presented in Supplementary Tables 14 and 15.

See new Fig 3 and supplementary tables 14 and 15.

Figure legend 3, line 888-902: **Fig. 3. Induced immune responses and immune cell constitution at discharge vs baseline, and correlation between immune cell subsets and induced immune responses.** a-b, Violin plots displaying induced concentration levels for (a) a subset of cytokines in response to LPS, R848, and CD3/CD28 at discharge ($n = 25$, except for CD3/CD28: $n = 24$) vs baseline ($n = 23$ except for CD3/CD28: $n = 21$), and (b) immune cell subsets at discharge ($n = 18$) vs baseline ($n = 28$). Solid dots represent individual patient measurements, colored by peak severity group; green = Grade 1, yellow = Grade 2, orange = Grade 3, red = Grade 4. Medians within each severity group at discharge and baseline are connected with a line, colored by peak severity. Data at baseline vs discharge were compared using the Wilcoxon signed-rank test, only including patients with paired baseline- and discharge samples available ($n = 18$ for induced immune responses, $n = 8$ for immune cell constitution). Adjustment for multiple testing was done using Bonferroni and adjusted $p < 0.05$ was considered statistically significant.

Supplementary table legend 14, line 58-61: **Supplementary table 14. Summary statistics from Wilcoxon signed-rank tests comparing cytokine concentrations at discharge vs. baseline.** Summary statistics for Wilcoxon signed-rank tests comparing TruCulture cytokine concentrations at discharge vs. baseline. Bonferroni adjustment is performed within each stimulus (9 tests).

Supplementary table legend 14, line 63-66: **Supplementary table 15. Summary statistics from Wilcoxon signed-rank tests comparing immune cell subset counts at discharge vs. baseline.** Summary statistics for Wilcoxon signed-rank tests comparing DuraClone immune cell subset counts at discharge vs. baseline. Bonferroni adjustment is performed for 8 tests.

4) In Figure 2 and 3, a statistical correlation between the immune populations present in each sample and the levels of cytokines should be performed. It is key to determine if levels of specific cytokines are linked to decreased or increased level of specific immune cells.

A: Thank you for addressing this relevant issue. We performed correlation analyses by Pearson's method for all stimulus cytokine variables and immune cell subsets at each timepoint (baseline, at/near peak, and discharge), adjusting for multiple testing within each timepoint using Bonferroni. The results are illustrated by three corrplots in new Fig 3C-E. The strongest correlations between cytokine variables and immune cell subsets were seen for LPS- and R848 stimulated IL-8 and monocytes, with significant correlations at all three timepoints. Several other LPS and R848-stimulated variables displayed significant correlations with monocyte- and neutrophil counts at baseline and at/near peak severity. However, none of the variables that we in this study found to be significantly associated with peak severity at baseline (LPS-stimulated IL-1B and R848-stimulated IL-12 and IL-17A) were significantly correlated with innate immune cell counts. Therefore we conclude that our findings of cytokine concentrations declining with increasing peak severity grade at baseline cannot alone be explained by changes in immune cell subset counts. We believe that adding these analyses further supports our previous findings and has contributed to a better manuscript.

Changes in manuscript:

Line 244-257: **Loss of correlation between immune cell subsets and their cytokine response as peak severity increases** Next, correlations between TruCulture immune responses and immune cells were investigated at baseline, at/near peak severity, and at discharge (Fig 3C-E). At baseline and at/near peak severity, monocytes and neutrophils displayed statistically significant positive correlations with LPS-stimulated IL-8. Statistically significant correlations were also observed between monocytes and R848-stimulated IL-1 β and TNF- α at baseline, and LPS- and R848 stimulated IL-12 at/near peak severity, while neutrophils displayed statistically significant correlations with R848-stimulated IL-10 at baseline and R848 stimulated IL-8 at/near peak severity. The only statistically significant correlation between immune cell counts and cytokine response at discharge was observed between monocytes and LPS-stimulated IL-8. Generally, the majority of significant correlations were observed at baseline, and occurred between the LPS and R848-stimulated cytokine responses (Fig 3C). Thus, the trends of LPS and R848-stimulated cytokine responses declining with increasing peak severity, and recovery of responses observed at discharge, do not seem to be explained by changes in immune cell counts, and therefore point toward functional changes.

Line 340-349: Intriguingly, we observed distinct trends of LPS- and R848 stimulated cytokine responses at baseline declining with increasing grade of subsequent peak disease severity, and identified LPS-stimulated IL-1B and IL-17A, and R848-stimulated IL-12, as individual baseline biomarkers significantly associated with subsequent peak severity. However, only a few LPS and R848-stimulated cytokine variables, such as IL-8, were significantly correlated with immune cell subset counts at baseline. Importantly, we did not observe a significant correlation between any of the LPS and R848-stimulated cytokine variables that displayed significant associations with peak severity and immune cell subset counts including monocytes, which represent an acknowledged source of LPS- and R848 stimulated cytokines.^{41,42} Thus, changes in immune cell counts cannot alone account for the pattern observed for LPS- and R848 stimulated cytokine levels.

Other points:

1) Lines 111-114: please acknowledge that regulation of IFN production is more complex, based on findings from Zhou Z. et al, 2020, Cell Host and Microbe; Lee JS. et al Sci. immunology 2020; Lucas C. et al Nature, 2020; Ziegler CGK et al, Cell 2021; Sposito B. et al Cell, 2021.

A: Thank you for pointing this out, we have rephrased this section and added the suggested references.

Changes in manuscript:

Line 113-120: Coherently, numerous studies highlight the presence of elevated circulating/plasma pro-inflammatory cytokines in patients with severe COVID-19,²²⁻²⁵ and high levels of interleukin (IL)-6 and tumor necrosis factor (TNF)- α in early disease correlate with severe disease trajectory and increased mortality.²² Type I interferon (IFN) responses have also been shown to play an important role in the pathogenesis underlying severe COVID-19, with several studies highlighting disturbances in the complex regulation of type I IFNs in different anatomical compartments as well as various stages of disease development.²⁴⁻³⁰ Correspondingly, presence of autoantibodies against type I IFNs were found enriched in patients with critical disease.³¹

2) It would be important to grade COVID-19 severity based on WHO parameters rather than with the 4 classes identified here. At least, describe the “conversion” to WHO severity as identified by WHO guidelines.

A: This is a really good point. We have addressed the main differences between our choice of severity grading system and the WHO severity classifications in the results and methods sections.

Changes (red text) in manuscript:

Line 148-154: Clinical severity during disease trajectories were mapped for each patient based on a day-by-day monitoring of levels of oxygen needed, need for treatment in the intensive care unit (ICU), and need for MV (Fig 1B-D). We defined 4 grades of disease severity (Fig 1C), modified from a previously published COVID-19 severity grading system.²⁴ The grading system differs from the World Health Organization's (WHO) severity classification³⁵ by being based primarily on interventions needed rather than diagnostic work-up/criteria (correspondence between the two grading systems is described in Supplementary Methods).

Online methods line 486-498: Based on the clinical disease trajectories, we defined a clinical severity scale with 4 grades of disease severity (Fig 1C), modified from a previously published COVID-19 severity grading system.²⁴ Where the World Health Organization (WHO) COVID-19 disease severity classifications³⁵ are based mainly on the symptoms of the patient, the severity scale applied here was based on needed interventions. Grade 1 was defined as requiring less than 3 liters per minute (L/min) of supplemental oxygen (O₂) to keep peripheral blood oxygen saturation (SAT) > 92%, and would correspond mainly to WHO classification *mild- and moderate disease*, however may also include milder cases of WHO classification *severe disease*. Grade 2 was defined as requiring ≥ 3 but <6 L/min of O₂ to maintain SAT > 92%, and would correspond mainly to WHO classification *severe disease*. Grade 3 was defined as requiring ≥ 6 L/min of O₂ and/or being admitted to the ICU, and would correspond to more severe cases of WHO classification *severe disease* as well as cases with *critical disease* not in need of mechanical ventilation. Grade 4 was defined as being treated with mechanical ventilator support, thus corresponding to cases with WHO classification *critical disease* in need of mechanical ventilation.

3) How is the normal reference level for each parameter analyzed in figure 2 determined?

A: This is described in the methods section.

See manuscript:

Online methods line 544-546: Reference intervals for all cytokine levels from all stimuli were based on TruCulture data from 31 healthy individuals and represent the range between the minimum and maximum cytokine concentration levels measured for each cytokine/stimulus.

METHODS

4) For figure 2C, it would be important to know the number of days each subjects took to get to disease peak.

A: We have added a column in Supplementary Table 2 showing days from hospital admission to collection of the peak severity sample.

5) Extended Data Figure 4, 5, 6 are not cited correctly in “Results” section.

A: Thank you for pointing this out, we have corrected this with correct references to the updated supplementary figures (new Supplementary Fig 5, 6, and 7) in the manuscript.

6) Figure 4 and Extended Data Figure 4, 5, 6. Clarify how the variable “Estimate” is calculated.

A: Thank you for highlighting that clarification is needed. The “Estimate” refers to the regression coefficient estimates of the variables in the OLS analyses. We have clarified this in the figure legends and in the methods section.

Changes in manuscript:

Online methods line 600-601: **Regression slopes and regression coefficient estimates were plotted with 95% confidence intervals** using the *jtools* package version 2.1.0.⁶¹

Figure legend 4, line 910-912: **Shaded areas behind regression lines represent 95% confidence intervals. Individual regression coefficient estimates for the cytokine variable and age are illustrated in a summary plot below each regression plot, hollow dots represent the estimates, bars represent 95% confidence intervals.**

Figure legend 4, line 917-918: **Regression coefficient estimates for the LPS and R848 stimulated cytokine variables and age at baseline vs recovery, hollow dots represent the estimates, bars represent 95% confidence intervals.**

Supplementary figure legend 5 (line 118-120), 6 (line 126-128) and 7 (139-141): **Collected regression coefficient estimates for the stimulated cytokine variables and age by all stimuli at baseline, hollow dots represent the estimates, bars represent 95% confidence intervals.**

Reviewer #2 (Remarks to the Author):

Svanberg et al. analyzed immune responses in 30 COVID-19 patients of varying disease severity using Trucount immune assay and flow cytometry. Their objective was to determine an early correlate that predicts disease severity. The study was well-designed and performed carefully; however, suffers from a lack of novelty. Several studies have shown impaired innate immune response, peripheral immune suppression despite enhanced cytokine response as a signature of severe COVID-19. The authors cite some but not all (for instance, Matthew et al. Science 2020) relevant studies. While the introduction has covered the most relevant details, the authors haven't cited appropriate literature. For e.g., Arunachalam et al. 2020 showed almost all that's described in lines 109-117 and the findings of the current study but has been cited somewhere in the discussion.

A: Thank you so much for this helpful feedback. Indeed, several important previous studies have highlighted impaired immune responses in COVID-19 patients and we have tried to cover the most relevant studies in the introduction of our paper, however we appreciate the opportunity to update the current manuscript by referencing the important studies mentioned above.

Matthew et al provides a thorough and in-depth characterization of T-cell and B-cell subsets in COVID-19 patients. Although we in our study assess T-cell receptor stimulated cytokine responses along with CD4+ and CD8+ T-cell subsets, our main findings are centered around the innate immune response. Therefore the study by Matthew et al falls outside of the main scope of the manuscript introduction, but we found it helpful to refer to the study when presenting and discussing our findings regarding the T-cell compartment in the “Results” and “Discussion” sections.

We appreciate the opportunity to improve our manuscript further by citing the important findings of Arunachalam and colleagues. The “Introduction” and “Discussion” sections have been updated to cite this study in a more relevant and clear manner. To highlight the novelty of our study, we also discuss the findings of our study in relation to the findings of Arunachalam *et al.* Similar to our study, Arunachalam *et al* also cover cytokine responses stimulated by TLR agonists in COVID-19 patients with different disease severity. However, there are some important differences between their findings and our study. The findings by Arunachalam *et al* are based on frozen PBMC samples, the study is cross-sectional with patient samples collected at different time-points without alignment, and the study does not demonstrate correlations between cytokine responses and severity grade. In contrast, our study is based on fresh collected patient samples that are analyzed in real-time to better reflect the *in vivo* conditions of the patient, our study presents longitudinal data where patient samples are collected at aligned timepoints, and our main finding demonstrates associations between impaired immune responses at time of hospital admission and *subsequently* developed peak severity, inferring identification of predictive biomarkers. Therefore, we believe that our study provides novel findings that have not been demonstrated in previous studies to our knowledge.

Changes in the manuscript:

Line 199-200 (citation to Matthew et al, reference #37): **Of note, a more in-depth characterization of the T- and B cell compartments in COVID-19 has been covered in a previous study.³⁷**

Line 362-365 (citation to Matthew et al, reference #37): **This was paralleled by restored immune cell levels, especially of CD8+ T cells, in coherence with previous studies highlighting the importance of adaptive T cell mediated responses for the clearance of SARS-CoV-2 infection.^{37,43,44}**

Line 112-119 (citation to Arunachalam et al, reference #25) : **Coherently, numerous studies highlight the presence of elevated circulating/plasma pro-inflammatory cytokines in patients with severe COVID-19,²²⁻²⁵ and high levels of interleukin (IL)-6 and tumor necrosis factor (TNF)- α in early disease correlate with severe disease trajectory and increased mortality.²² Type I interferon (IFN) responses have also been shown to play an important role in the pathogenesis underlying severe COVID-19, with several studies highlighting disturbances in the complex regulation of type I IFNs in different anatomical compartments as well as various stages of disease development.²⁴⁻³⁰ Correspondingly, presence of autoantibodies against type I IFNs were found enriched in patients with critical disease.³¹**

Line 352-358 (discussing findings of Arunachalam et al, reference #25): **Suppressed expression of cytokines like TNF- α by innate immune cells in response to viral and bacterial TLR stimulation in COVID-19 patients compared with healthy individuals has previously been reported.²⁵ These findings were based on samples collected during ongoing disease, thus furthering the findings of this study. Thus, this is to our knowledge the first study demonstrating early impaired innate immune responses based on TLR-stimulation of fresh whole-blood samples collected from COVID-19 patients at time of hospitalization, that furthermore shows an association with subsequently developed disease severity.**

Fig.2-3: Are any of the differences statistically significant? Please provide statistical analyses.

A: Thank you for highlighting the lack of statistical analyses in Figs 2-3. This was also addressed by Reviewer #1, therefore we refer to the response given to Reviewer #1 above.

The argument that these signatures could be identified at the time of hospital admission, and therefore is predictive of the disease trajectory is valuable, but such a claim needs validation. Could the authors test their signature in an additional set of patients?

A: This is a valid and relevant point, and we appreciate the challenge to push our study further. We selected an additional set of 20 patients for the validation cohort (5 patients in each of the four peak severity groups) who were enrolled in the COVIMUN study as early as possible following the original cohort (on which the findings of this study is based). The patients in the validation cohort were also

selected to match the original cohort on the most important baseline characteristics. The LPS and R848 stimulated TruCulture cytokine concentration data from the validation cohort were then used to test the predictive performance of our immune response signature (results are described and discussed below). We believe that adding this validation of the signature has provided a more complete story, and has significantly improved the manuscript.

Changes in manuscript:

See new supplementary table 9.

Supplementary table legend 9, line 36: **Supplementary table 9. Baseline patient characteristics of the validation vs the original cohort.**

Line 305-318: **Validation of the baseline stimulation signature in a separate cohort of hospitalized COVID-19 patients** Next, we tested the signature identified by the LPS+R848 model in a separate cohort. We selected 20 patients with five patients in each peak severity group. The patients in the validation cohort were enrolled in the COVIMUN study in the months following the original cohort (October 2020-January 2021). Patient baseline characteristics of the validation cohort were comparable to the original cohort (Supplementary Table 9), except having a lower proportion of patients with immune dysfunction (30% vs 57%). We found that although the signature was not very predictive of specific peak severity grade vs the rest, it performed best in predicting the two highest severity grades vs the two lowest (sensitivity=0.7, specificity=0.8, diagnostic odds ratio=9.3, Matthews correlation coefficient=0.5, Fig 6A, Supplementary Fig 10). Coherently, the LPS- and R848 induced cytokine responses displayed the same visual trend of declining values with increasing peak severity grade as observed for the original cohort (Supplementary Fig 6B). Thus, the immune signature based on LPS+R848 stimulated responses represents a model with explanatory as well as predictive value which may improve our understanding of early functional immune impairments entailing risk of severe COVID-19.

Line 386-402: Importantly, the LPS+R848 model was superior to the model based on only the three significant variables identified in the individual OLS analyses. This supports that the lack of additional significant linear relationships between individual CSR variables and severity was likely a power issue. Given the small cohort size (n=23), large number of predictors (18), and high r^2 (0.91), we expected the LPS+R848 model to be overfitted to this specific cohort. This was in part reflected in the lower accuracy of the model to predict one specific peak severity grade vs the rest when tested on the validation cohort. The presence of two pronounced outliers, uneven distribution of patients in peak severity groups in the original cohort, along with a larger proportion of subjects with immune dysfunction in the original cohort, likely contributes further to the performance. Nevertheless, when reducing the predicted outcome to two instead of four severity grades (grade 3-4 vs grade 1-2), the model identified 7 out of 10 patients with risk of needing >6L of O₂, admission to ICU, need for MV, or death. From a clinical perspective, this could provide valuable information allowing for closer monitoring and earlier interventions. While supporting the predictive value of the LPS- and R848 based immune signature, the results from the validation cohort highlight that further development and optimization of this model is warranted. Exploring the signature in a larger cohort, adjusting the model output limits defining a specific peak severity grade, and investigating threshold levels of cytokine concentrations constitute a few parameters that could be optimized to improve the accuracy and performance.

Online methods line 446-450: Thirty patients enrolled in the COVIMUN study at three hospitals were included in the original cohort used for this study, and a validation cohort of twenty patients were selected for validation of our immune response signature. Patients in the original cohort were enrolled between April 19th 2020 and October 14th 2020, and patients in the validation cohort were enrolled between October 14th 2020 and January 3rd 2021.

Online methods line 632-640: For validation of the best model, predictions were performed based on baseline TruCulture cytokine concentration data (standardized and log-transformed) from the validation cohort (n=20). We predefined the thresholds for categorizing the (continuous) model output as follows; <1.5: Severity group 1, ≥1.5<2.5: Severity group 2, ≥2.5<3.5: Severity group 3, and ≥3.5: Severity group 4. Sensitivity/recall, specificity, false positive rate, false negative rate, positive predictive value/precision, negative predictive value, false discovery rate, false omission rate,

positive likelihood ratio, negative likelihood ratio, diagnostic odds ratio, and Mathew' s correlation coefficient were calculated for predicting severity grade 3 or 4 vs grade 1 or 2 and vice versa, as well as severity grade 1 alone and severity grade 4 alone (Fig 6A, Supplementary Fig 10A-D).

See new figures added: Fig 6 and Supplementary Fig 10.

Figure legend 6, line 936-948: **Fig. 6. Validation of the immune response signature in a separate cohort.** The LPS+R848 model was validated on a separate cohort of hospitalized COVID-19 patients (n=20, 5 in each peak severity group). **a**, Sensitivity/recall, specificity, false positive rate (FPR), false negative rate (FNR), positive predictive value (PPV) /precision, negative predictive value (NPV), false discovery rate (FDR), false omission rate (FOR), diagnostic odds ratio (DOR), and Mathew' s correlation coefficient (MCC) are presented for predicting severity grade 1 alone, grade 1-2, grade 3-4, and grade 4 alone. **b**, Induced cytokine levels in response to LPS and R848 based on baseline data from the validation cohort (n=20). Patients are grouped based on future peak severity: Grade 1 (n=5, green), Grade 2 (n=5, yellow), Grade 3 (n=5, orange), Grade 4 (n=5, red). Box edges represent the 25th and 75th percentiles, whiskers extend towards the most extreme values but no further than +/- 1.5 times the interquartile range from the hinge. Hollow dots beyond whiskers represent outliers. Solid dots represent individual measurements. Blue shaded areas represent the normal reference interval. Cytokine concentration levels and immune cell subset counts are presented on a log₁₀ y-axis.

Supplementary figure legend 10, line 169-176: **Supplementary Fig. 10. Validation of the immune response signature in a separate cohort.** Predictions were performed using the LPS+R848 model based on baseline TruCulture cytokine concentration data (standardized and log-transformed) from a separate validation cohort of hospitalized COVID-19 patients (n=20, 5 in each peak severity group). **a-d**, Calculations of sensitivity/recall, specificity, false positive rate, false negative rate, positive predictive value/precision, negative predictive value, false discovery rate, false omission rate, positive likelihood ratio, negative likelihood ratio, diagnostic odds ratio, and Mathew' s correlation coefficient are presented for predicting (a) severity grade 3-4 vs 1-2, (b) severity grade 1-2 vs 3-4, (c) severity grade 1 vs 2-4, and (d) severity grade 4 vs 1-3.

Reviewer #3 (Remarks to the Author):

This article shows that the potential to predict the outcome of the COVID-19 severity using whole blood samples at the early stage of the course of symptom (at the time of hospital admission). The results and conclusions from the study show the novel information that helps decision making for those who work at clinical sites. The overall manuscript is well written, and the methodologies (including statistical analysis) were properly performed. Two statistical models were built for clinical data showing a similar trend result, which strengthen the hypothesis of this study. This study analysed pilot data with small samples, but the results obtained from this study contains crucial evidence to encourage further study in the future applying wider data. Several points that improve the validity of this study and understanding for readers, which I believe, are the following:

A: Thank you for the valuable and constructive feedback, and for the excellent suggestions to help us improve the manuscript.

1. There are several approaches to select covariates for prediction models. The methods applied here [based on statistical significance for the OLS model (age) and based on a priori knowledge for potential confounders for the OLS with lasso model (6 variables)] are widely used approaches. However, why the inconsistent selection criteria were used for both methods was not explained.

Were the criteria of covariates selection designed before the analysis?

Is the reason for selecting one covariate (i.e., age) for the OLS model also due to the small sample size?

The different trend between the two models shown in this study (lines 329–331) might be influenced by these confounder adjustments. Please consider explaining why the different selection criteria were used for the models in the method section and discuss potential/expected bias from the adjustments.

A: Before conducting the OLS analyses we decided to test a selection of covariates that we expected to have impact on peak severity in univariate analyses as described in Online methods (line 587-594) and presented in Supplementary table 3. Due to the lack of any significant impact of these covariates on peak severity, they were not included in the OLS to maximize power. We decided to include “Age” nevertheless, as it is such a well-established risk factor for COVID-19 disease severity.

Originally, having identified three statistically significant variables at baseline associated with peak severity, we proceeded to explore the predictive potential of the LPS- and R848 stimulated cytokine concentration data at baseline alone in the LASSO regression analyses, and therefore the variable “Age” was not included for these analyses.

However, to address the raised issue regarding the discrepancy between the two different models (OLS and LASSO), we now demonstrate the impact of “Age” in the LASSO regression model by including it within in each bin (illustrations of this are added in Supplementary Fig 8). We show that “Age” generally had either no impact or slightly improved fit for a few bins with low r^2 . Notably, inclusion reduced r^2 for best performing bins, hereunder LPS+R848. Of all bins tested with or without “Age”, LPS+R848 without “Age” still performed the best. Altogether, this indicates that “Age” likely serves as a proxy for the LPS and R848 stimulated responses. Therefore, expressly for the purposes of determining a cytokine signature, we chose to keep the model based on LPS+R848 and without “Age” in the main manuscript.

In addition, the LPS+R848 model demonstrated a high performance while keeping all but 4 variables. Therefore, this model also provides explanatory value helping us to better understand the underlying biology that likely is determinant for different COVID-19 disease severity courses.

We have addressed the impact of “Age”, and the choice of including/excluding the variable in the different models in the manuscript.

Changes in manuscript:

Line 263-268: Using univariate linear models, we found no significant relationship between peak severity and clinical covariates (Supplementary Table 3). While this suggested little to no confounding in our experimental design, we decided to nevertheless adjust for age in all CSR effect size estimates given its well-established risk for severe COVID-19,¹⁶ and as it was the covariate closest to demonstrating an impact ($p=0.08$). **Given our small sample size, the other covariates were excluded to maximize statistical power.**

Line 286-292: We then used OLS in combination with lasso penalties to assess $CSR_{baseline}$ variables in bins of stimuli to identify the subset of predictive variables that were most likely to generalize our model while limiting potential for overfitting (Supplementary Fig 8C-E, bins are described in online methods). **We focused on $CSR_{baseline}$ as the only endogenous variables to maximize power as we had already confirmed the absence of possible confounding by other exogenous factors in univariate analyses (Supplementary Table 3). Specifically, *age* was not included as its level of significance suggests its inclusion would be counterproductive.**

Line 378-382: **Including age to the bins had little to no impact, reduced r^2 for the best performing models, and improved r^2 for poor performing models. Given that age is a well-established risk factor for severe COVID-19, this sensitivity analysis indicates that it likely serves as a proxy for more the precise signals represented by LPS- and R848 induced immune responses in our patient cohort. However, this warrants further investigation in larger cohorts.**

See new supplementary figure 8G-H.

Supplementary figure legend 8, line 151-155: **g**, R^2 from all bins tested with and without including "Age" as a variable; red bars represent models excluding "Age", blue bars represent models including "Age". **h**, Best lambda from all bins tested with and without including "Age" as a variable; red bars represent models excluding "Age", blue bars represent models including "Age".

2. The same as the above issue. Lines 329–331: this could be (could be not though) because of not controlling confounders other than age. Please discuss the limitation if it is difficult to adjust confounders (e.g. due to small sample size or other reasons).

A: As we understand it, the issue raised here is regarding the Poly:IC stimulated responses, and how the individual variables did not show a strong association with peak severity in the OLS analyses adjusting for "Age", while the Poly:IC stimulated responses in the LASSO regression analysis not including "Age" as a variable, provided the second best performing model. This is definitely a relevant and interesting point.

In the OLS for Poly:IC stimulated variables, we can see from the regression coefficient estimates (Supplementary Fig 5 F) that the "Age" variable has an impact, but not significant. Meanwhile the impact of "Age" was bigger on the OLS from the other stimuli. In the LASSO regression analysis for Poly:IC, performance was the highest of all single-stimuli bins, and when "Age" was added, performance was slightly improved (Supplementary Fig 10G). As mentioned in the prior answer, we believe the *weak* signal from "Age" is likely due to hidden correlation (i.e. proxied) with other stimuli. In this sense, it would be improper to include it under this null hypothesis. The same could be true for other exogenous signals, however the point is mostly moot in that these other variables also displayed poor predictive power when tested alone.

Therefore, we think that the discrepancy between the results from OLS and LASSO for Poly:IC is not due to adjusting or not adjusting for the "Age" variable. Since the other potential clinical confounders displayed a much weaker association with peak severity in univariate analyses (Supplementary table 3), we do not think that they would have made an impact if included in either analysis.

As described in the previous response, we have tried to make it more clear in the manuscript that we decided not to control for more variables in either analysis due to the small sample size.

3. Line 234, Fig.3D: there is no Fig. 3D panel. Is it Fig.3A?

A: Thank you for pointing this out. This has been corrected in the manuscript.

4. Lines 278-280, "This suggests that despite ... when combined.": can you show the evidence of this significant orthogonal effect from the analysis not only from the difference of r^2 ? If yes, please add the analysis. If not, please move this interpretation to the discussion section only (together with lines 331–336).

A: We agree that this statement needed to be backed up by further evidence. We performed PCA projecting the TruCulture data from all stimuli. The angle between the loadings of LPS and R848 suggests a synergy between the stimuli (Supplementary fig 8B). We have also rephrased this section in "Results", see below.

Changes in manuscript:

Line 296-300: Furthermore, the LPS+R848 model performed better than the model based on only the three statistically significant variables (LPS-stimulated IL-1 β , R848-stimulated IL-12 and IL-17A) from the individual OLS analyses ($r^2=0.48$). This suggests that the correlation between R848 and LPS responses synergistically enhances the predictive

value of our model when combined. This is further supported by the relationship between R848 and LPS data projections in PCA (Supplementary Fig 8B).

See supplementary figure 8B.

Supplementary figure legend 8B, line 146-148: **b**, Principal component analysis displaying TruCulture data from all five stimuli at baseline. Arrows extend from origin to centroid of each stimulus representing the loadings of each stimuli in the projected dimensions.

5. Lines 312–316: how evenly the patients (with dexamethasone-treatment/immunosuppressive-conditions/comorbidities) were distributed was not explained. Proportionally even or the absolute numbers of them were even among groups? It is good for readers to show the proportion of each condition.

A: Thank you for pointing out that this was not clear. We have clarified in the manuscript that distribution is proportionally even among groups, and refer to the visualization of the proportion of each condition in the four groups in Supplementary figure 1D-F and Supplementary table 2.

Changes in manuscript:

Line 164-171: These patients were **proportionally** evenly distributed across the four peak severity groups (Supplementary Fig 1D). Twenty-six patients (87%) had at least one comorbidity, and eleven patients (37%) had two or more comorbidities. Comorbidities were also **proportionally** evenly distributed across peak severity groups (Supplementary Fig 1E-F). Dexamethasone treatment was not approved as standard of care at the time of the pandemic where most patients in our study were included,³⁶ therefore only nine patients received dexamethasone throughout the study of which six had it administered prior to baseline sampling. They were also **proportionally** evenly distributed across peak severity groups (Supplementary Table 2).

6. Lines 452–460: do authors consider that not all missingness were at random (just some of them were mentioned as missing at random)? How they were missing and how this missingness influenced the result should be discussed.

A: Thank you for making us aware of this. We do consider all missingness were at random, and this has been clarified in methods.

Online methods line 513-516: Seven patients were included in the study >7 days after initial hospitalization **for different (random) reasons, and thus, baseline samples for both TruCulture and DuraClone were missing for these patients** (four patients from Peak Severity Grade 1, three patients from Peak Severity Grade 3).

7. Lines 522–524: the principal component analysis did not show the same trend as Isomap analysis, but it is good to show the figure of PCA result (the similar type of the plot of Isomap analysis) in the supplementary material.

A: We have added the figure of the PCA, see Supplementary Fig 8A.

Supplementary figure legend 8, line 145: **a**, Principal component analysis displaying severity levels based on LPS+R848 TruCulture baseline data

8. Lines 541–543: does “r2” stand for the “coefficient of determination”? Please define in the method part. Is the equation (line 543) properly shown (or typo)? It seems that the denominator and numerator are opposite. Also, since the predictions were made for individual-level, the notation of the summation should be used.

A: Thank you for making us aware of these remarks. R^2 stands for “coefficient of determination”, this has been updated in “Methods”. The equation was inverted by mistake, thank you for pointing this out, it has now been corrected.

Changes in manuscript:

Online methods line 627-632: Predictions were then performed on the original cohort using the subset of variables on which the best model was based. r^2 (coefficient of determination) for each bin (Supplementary Fig 8C) was calculated as follows:

$$r^2 = 1 - (\Sigma(\text{Severity_Predicted} - \text{Severity})^2 / \Sigma(\text{Severity} - \text{mean}(\text{Severity}))^2)$$

Where “Severity” is the peak severity grade indicated categorically with values 1-4, and “Severity_Predicted” is the predicted peak severity based on the given model.

REVIEWERS' COMMENTS:

Reviewer #1 (Remarks to the Author):

The new version of the manuscript has been highly improved and although many parameters only show a trend that is not statistically significant, the identification of altered IL1/TNF production at the baseline upon LPS stimulation, and/or LPS+R848 stimulus combination may be used to predict COVID severity.

I have only minor comments:

- 1) Please delete “visual trend” and only leave “trend”
- 2) Please in Lines 115-119 refer to “Type I and type III interferons”

Reviewer #2 (Remarks to the Author):

The authors addressed all my concerns. The manuscript has been significantly improved.

Reviewer #3 (Remarks to the Author):

The manuscript has significantly improved with an additional validation for the prediction. All comments from this reviewer were properly addressed. Especially, the methodologies of confounding adjustment, which I pointed out, were clearly mentioned. Now I understood that the purpose of LASSO regressions was to explore the predictive performance of stimulated cytokine concentration data at baseline alone.

At first, I misunderstood that the OLS with LASSO penalties controlled potential confounders. From Figure 5A, I misunderstood sex, age, and immune suppression were adjusted as confounders in the models because there were rows of them. But they were just shown in the figure for the readers' information, and with the revised explanations, it is clear that they were not included in the models.

The comments below are not directly related to the scope of the current study, but for future studies.

Although the inclusion of “age” in the models did not improve prediction performance, the sensitivity analysis including “age” as a confounder had important insights because of the following reasons.

First, “age likely serves as a proxy for more the precise signals represented by LPS- and R848 induced immune responses in our patient cohort.” (lines 380-381), this means, I think, “age” might be a potential confounder, and it is important to include it in the model when you want to estimate the causal effect (not prediction) of LPS and R848 stimulated responses on peak severity excluding the effect of age (But this is kind of out of scope from this study. For prediction purposes, exploring the best prediction model, referring r^2 and λ is a right way.) Second, the impact of Poly:IC stimulated cytokine on peak severity was emphasized by adjusting age (Supplementary Fig 8G, H). Poly:IC stimulated cytokine might have a good potential for both prediction performance and a causal effect.

For future studies, it must be interesting to investigate the causal impact of Poly:IC stimuli responses at baseline on the severity, which may be not a proxy of “age”, and to explore prediction ability on

peak severity in a larger cohort.

Response to Reviewers' Comments (author comments in blue):

To the Reviewers,

We are very grateful for the positive response and the final feedback provided by the three reviewers.

We have complied with the last suggestions and believe that the manuscript now is suitable for publication in its current version.

For simplicity, the reviewer comments/questions are in bold and our response is in blue and red (the latter when based on novel data included in the manuscript).

Best Regards,

Rebecka on behalf of all authors

Reviewer #1 (Remarks to the Author):

The new version of the manuscript has been highly improved and although many parameters only show a trend that is not statistically significant, the identification of altered IL1/TNF production at the baseline upon LPS stimulation, and/or LPS+R848 stimulus combination may be used to predict COVID severity.

I have only minor comments:

- 1) Please delete "visual trend" and only leave "trend"
- 2) Please in Lines 115-119 refer to "Type I and type III interferons"

A: Thank you for all the feedback and helping us improve the manuscript. We have changed the wordings according to the above mentioned suggestions in the final manuscript.

Reviewer #2 (Remarks to the Author):

The authors addressed all my concerns. The manuscript has been significantly improved.

A: We thank you for the valuable feedback that helped us improve the manuscript.

Reviewer #3 (Remarks to the Author):

The manuscript has significantly improved with an additional validation for the prediction. All comments from this reviewer were properly addressed. Especially, the methodologies of confounding adjustment, which I pointed out, were clearly mentioned. Now I understood that the purpose of LASSO regressions was to explore the predictive performance of stimulated cytokine concentration data at baseline alone.

At first, I misunderstood that the OLS with LASSO penalties controlled potential confounders. From Figure 5A, I misunderstood sex, age, and immune suppression were adjusted as confounders in the models because there were rows of them. But they were just shown in the figure for the readers' information, and with the revised explanations, it is clear that they were not included in the models.

The comments below are not directly related to the scope of the current study, but for future

studies.

Although the inclusion of “age” in the models did not improve prediction performance, the sensitivity analysis including “age” as a confounder had important insights because of the following reasons.

First, “age likely serves as a proxy for more the precise signals represented by LPS- and R848 induced immune responses in our patient cohort.” (lines 380-381), this means, I think, “age” might be a potential confounder, and it is important to include it in the model when you want to estimate the causal effect (not prediction) of LPS and R848 stimulated responses on peak severity excluding the effect of age (But this is kind of out of scope from this study. For prediction purposes, exploring the best prediction model, referring r^2 and λ is a right way.) Second, the impact of Poly:IC stimulated cytokine on peak severity was emphasized by adjusting age (Supplementary Fig 8G, H). Poly:IC stimulated cytokine might have a good potential for both prediction performance and a causal effect.

For future studies, it must be interesting to investigate the causal impact of Poly:IC stimuli responses at baseline on the severity, which may be not a proxy of “age”, and to explore prediction ability on peak severity in a larger cohort.

A: We are very happy to hear that our revised manuscript has clarified the concerns raised previously by Reviewer #3. We further thank you for the comments provided here, they will indeed be helpful in furthering the findings of this study as we proceed to explore the predictive potential of these induced immune responses as well as their causal/explanatory value in future studies.